# Phosphorylation of P-stalk proteins defines the ribosomal state for interaction with auxiliary protein factors

Kamil Filipek [ID][1,13], Sandra Blanchet [ID][2,11,13], Eliza Molestak[1], Monika Zaciura[1], Colin Chih-Chien Wu[3,4], Patrycja Horbowicz-Drożdżal[1], Przemysław Grela [ID][1], Mateusz Zalewski [ID][5], Sebastian Kmiecik[5], Alan González-Ibarra[1], Dawid Krokowski[1], Przemysław Latoch [ID][6], Agata L Starosta [ID][6], Mateusz Mołoń [ID][7], Yutian Shao [ID][1], Lidia Borkiewicz [ID][1,12], Barbara Michalec-Wawiórka[1], Leszek Wawiórka[1], Konrad Kubiński [ID][8], Katarzyna Socała [ID][9], Piotr Wlaź [ID][9], Kyle W Cunningham [ID][10✉], Rachel Green [ID][3✉], Marina V Rodnina [ID][2✉] & Marek Tchórzewski [ID][1✉]

## Abstract

Ribosomal action is facilitated by the orchestrated work of *trans*-acting factors and ribosomal elements, which are subject to regulatory events, often involving phosphorylation. One such element is the ribosomal P-stalk, which plays a dual function: it activates translational GTPases, which support basic ribosomal functions, and interacts with the Gcn2 kinase, linking the ribosomes to the ISR pathway. We show that P-stalk proteins, which form a pentamer, exist in the cell exclusively in a phosphorylated state at five C-terminal domains (CTDs), ensuring optimal translation (speed and accuracy) and may play a role in the timely regulation of the Gcn2-dependent stress response. Phosphorylation of the CTD induces a structural transition from a collapsed to a coil-like structure, and the CTD gains conformational freedom, allowing specific but transient binding to various protein partners, optimizing the ribosome action. The report reveals a unique feature of the P-stalk proteins, indicating that, unlike most ribosomal proteins, which are regulated by phosphorylation in an on/off manner, the P-stalk proteins exist in a constantly phosphorylated state, which optimizes their interaction with auxiliary factors.

**Keywords** Ribosomal Proteins; Ribosomal Stalk; Ribosome; Phosphorylation; Gcn2 Kinase
**Subject Categories** Post-translational Modifications & Proteolysis; Translation & Protein Quality

## Introduction

Protein synthesis is a highly energy-consuming process where the ribosome acting as a Brownian machine harnesses thermal-structural fluctuations into directed motion on mRNA (Frank and Gonzalez, 2010; Rodnina et al, 1997; Wilson and Noller, 1998). The unidirectional trajectory for the translational apparatus is conferred by GTP hydrolysis catalyzed by auxiliary translational GTPases (trGTPases). The eukaryotic ribosomal P-stalk (and its analog L7/12 in bacteria) plays a pivotal role in the recruitment and activation of trGTPase to the ribosome, which leads to subsequent structural rearrangements, driving mRNA decoding and the ribosome movement on the mRNA (Liljas and Sanyal, 2018; Mohr et al, 2002; Savelsbergh et al, 2000). Recent findings show that, the eukaryotic P-stalk is also responsible for the activation of the stress-related kinase Gcn2, which couples the translational machinery with the Integrated Stress Response (ISR) pathway (Gupta and Hinnebusch, 2023; Harding et al, 2019; Inglis et al, 2019; Masson, 2019). Thus, the ribosome is not only responsible for protein biosynthesis per se, but has also been recognized as a sentinel for stress sensing able to trigger ISR (Kim and Zaher, 2022). In this respect, the ribosomal P-stalk appears as a key player in both

[1]Department of Molecular Biology, Institute of Biological Sciences, Maria Curie-Skłodowska University, Lublin, Poland. [2]Department of Physical Biochemistry, Max Planck Institute for Multidisciplinary Sciences, Göttingen, Germany. [3]Department of Molecular Biology and Genetics, Howard Hughes Medical Institute, Johns Hopkins University School of Medicine, Baltimore, MD, USA. [4]Section of Translational Control of Gene Expression, RNA Biology Laboratory, Center for Cancer Research, National Cancer Institute, National Institutes of Health, Frederick, MD, USA. [5]Biological and Chemical Research Center, Faculty of Chemistry, University of Warsaw, Warsaw, Poland. [6]Institute of Biochemistry and Biophysics, Polish Academy of Sciences, Warsaw, Poland. [7]Institute of Biology, University of Rzeszow, Rzeszow, Poland. [8]Department of Molecular Biology, Institute of Biological Sciences, John Paul II Catholic University of Lublin, Lublin, Poland. [9]Department of Animal Physiology and Pharmacology, Institute of Biological Sciences, Maria Curie-Skłodowska University, Lublin, Poland. [10]Department of Biology, Johns Hopkins University, Baltimore, MD, USA. [11]Present address: Institute for Integrative Biology of the Cell, I2BC, CEA, CNRS, Université Paris-Saclay, Gif-sur-Yvette, France. [12]Present address: Department of Biochemistry and Molecular Biology, Medical University of Lublin, Aleje Racławickie 1, 20-059 Lublin, Poland. [13]These authors contributed equally: Kamil Filipek, Sandra Blanchet. ✉E-mail: kwc@jhu.edu; ragreen@jhmi.edu; rodnina@mpinat.mpg.de; marek.tchorzewski@mail.umcs.pl

ribosomal activities, displaying dual functionality and being involved in the interplay with trGTPases and Gcn2 kinase. The overall architecture of the stalk is similar in all domains of life: the stalk is composed exclusively of ribosomal proteins, the uL11 and the heterocomplex composed of uL10 in a complex with lateral bL12 in bacteria and P1-P2 dimers in eukaryotes. The uL11 and uL10 proteins belong to the conserved ribosomal proteins found in all domains of life and form the base of the stalk by interacting with domain IV of rRNA (Liljas and Sanyal, 2018). In Archaea/Eukarya clades, the P-stalk forms heptamers uL10(aP1)$_6$ or pentamers uL10(P1-P2)$_2$, respectively, whereas the bL12-stalk in bacteria has a bimodal distribution between two or three lateral bL12 dimers, uL10(bL12)$_{4(6)}$, except for 8 bL12 copies in Cyanobacteria(Davydov et al, 2013). Despite the overall structural similarity, the stalk complexes are not functionally exchangeable between organisms from the three life domains, being compatible only with homologous trGTPases (Mochizuki et al, 2012). Such specificity is conferred by the Eukarya-specific domain II of the uL10 protein (Mochizuki et al, 2012) along with uL11 (Uchiumi et al, 2002), and by the lateral stalk elements bL12 and P1-P2/aP1 for bacterial and archaeal/eukaryotic ribosomes, respectively (Mochizuki et al, 2012). The uL11 forms a separate stalk entity and represents an element shaping the architecture of the stalk (Wawiorka et al, 2016). Functionally, uL11, together with uL10, serves as an anchoring point for trGTPases. As shown in bacterial ribosomes, uL11 can form an Arc-like connection with the GTPase domain of trGTPases (Datta et al, 2005), while in the eukaryotic ribosome, uL11 interacts with domain II of uL10, which can stabilize the translation factors on the ribosome (Shao et al, 2022; Yang et al, 2022). On the other hand, the uL10 anchor the lateral stalk proteins bL12 or P1-P2/aP1, which form protruding structural element. Regardless of their origin, the bL12 or P1-P2/aP1 proteins display characteristic topology with the N-terminal domain (NTD) and the C-terminal domain (CTD) connected by a flexible linker region (Bernado et al, 2010; Diaconu et al, 2005; Grela et al, 2007; Lee et al, 2012). Both NTDs and CTDs are evolutionarily conserved within each domain of life, but not across, showing that the lateral proteins should be regarded as analogous (Grela et al, 2008a). The NTDs of lateral stalk proteins form a compact dimeric structure, which is responsible for the interaction with the helical regions of the uL10 scaffold (Diaconu et al, 2005; Grela et al, 2012; Lee et al, 2012). The striking structural difference between bacterial and eukaryotic lateral proteins refers to their CTDs. Even though CTDs in bacteria and eukaryotes are critical for the recruitment of trGTPases, they are neither evolutionarily nor structurally related (Grela et al, 2008a). The bacterial CTD is a globular domain (Diaconu et al, 2005; Leijonmarck and Liljas, 1987) involved in binding of trGTPases and facilitating GTP hydrolysis and stabilization of trGTPase binding to the ribosome (Diaconu et al, 2005); whereas the archaeal/eukaryotic CTD lacks a defined structural organization (Lee et al, 2012), displaying properties of an intrinsically disordered protein (IDP) (Mishra and Hosur, 2015). The eukaryotic CTD function has not been resolved as yet; however, the archaeal CTD is directly involved in binding of various trGTPases, including eIF5B (Murakami et al, 2018), eEF1A (Nomura et al, 2012), eEF2 (Tanzawa et al, 2018), and the ABCE1 recycling factor (Imai et al, 2018). The most unique feature of the P-stalk protein CTDs is the highly conserved amino acid motif EESEESDDDMGFGLFD (Tchorzewski, 2002), which is almost identical for uL10, P1, and P2. The sequence is attributed to IDPs, as supported by structural analyses (Grela et al, 2007; Lee et al, 2012); however, as shown for the archaeal CTD, it can also adopt a helical conformation upon trGTPase binding (Murakami et al, 2018; Nomura et al, 2012; Tanzawa et al, 2018).

All eukaryotic P-stalk proteins, uL10, P1, and P2, are specific substrates for the CK2 protein kinase, which can modify two serine residues in the SEESDDD motif in the CTD (Hasler et al, 1991; Zambrano et al, 1997; Zinker and Warner, 1976). While the first serine in the motif is found mostly in mammals, the second serine residue in the motif is highly conserved across all eukaryotes, with only a few notable exceptions, such as pathogenic protists *Trypanosoma* sp. and *Plasmodium* sp., where serine is substituted with glutamic acid. In Archaea, such as extremophilic *Haloarcula morismortui*, the serine residue is substituted with glutamic residue as well (Tchorzewski, 2002) (Fig. 1). Importantly, although the phosphorylation of P-stalk proteins was discovered more than five decades ago, the phospho-status of the stalk proteins and the physiological role of this modification has not been revealed. The only observation was that phosphorylation of the P1/P2 proteins is not required for ribosome binding, but P2 phosphorylation can promote eEF2 binding (Vard et al, 1997). Protein phosphorylation represents an ubiquitous molecular strategy used by the cell to regulate metabolic activity, affecting the conformation of the proteins and, at the same time, influencing numerous aspects of protein functions (Nishi et al, 2011). Reversible phosphorylation and dephosphorylation can drive structural transitions in the IDPs (Iakoucheva et al, 2004); for example, the CK2 kinase plays a role in regulating the structural transitions of various IDPs and affects the balance between their aggregation and disaggregation as well as their binding to other proteins, DNA, and RNA (Zhang et al, 2020). It is well established that the phosphorylation of proteins involved in all major steps of translation, including translation factors, regulates the performance of the translational machinery and, consequently, the cellular proteome (Simsek and Barna, 2017). The most prominent example is the eIF2α phosphorylation by the Gcn2 kinase upon ISR pathway induction, which blocks the translation of house-keeping mRNAs and stimulates the expression of mRNAs involved in stress adaptation (Masson, 2019). Another example is the phosphorylation of the 4E-BP protein family. In its non-phosphorylated state, 4E-BP binds to eIF4E, and this binding prevents efficient initiation of translation. Conversely, phosphorylation of 4E-BP by mTORC1 kinase disrupts its interaction with eIF4E, thereby stimulating the initiation of translation (Bah et al, 2015). In addition, several ribosomal proteins undergo phosphorylation, including S6, which was discovered as the first ribosomal protein to be phosphorylated (Gressner and Wool, 1974) and its phosphorylation promotes initiation of translation (Bohlen et al, 2021). In turn, phosphorylation of uL13 governs its association with the ribosome (Mazumder et al, 2003).

In this study, we address the long-standing unresolved question of phosphorylation in eukaryotic ribosomal P-stalk proteins to understand the biological role of this post-translational modification. We show that P-stalk proteins in cells predominantly exist in a phosphorylated state. This post-translational modification lowers the interactions with trGTPases, leading to an optimal translation rate, particularly during the decoding step. Furthermore, phosphorylated P-stalk proteins may play an important role in the Gcn2-dependent stress response by facilitating ribosome

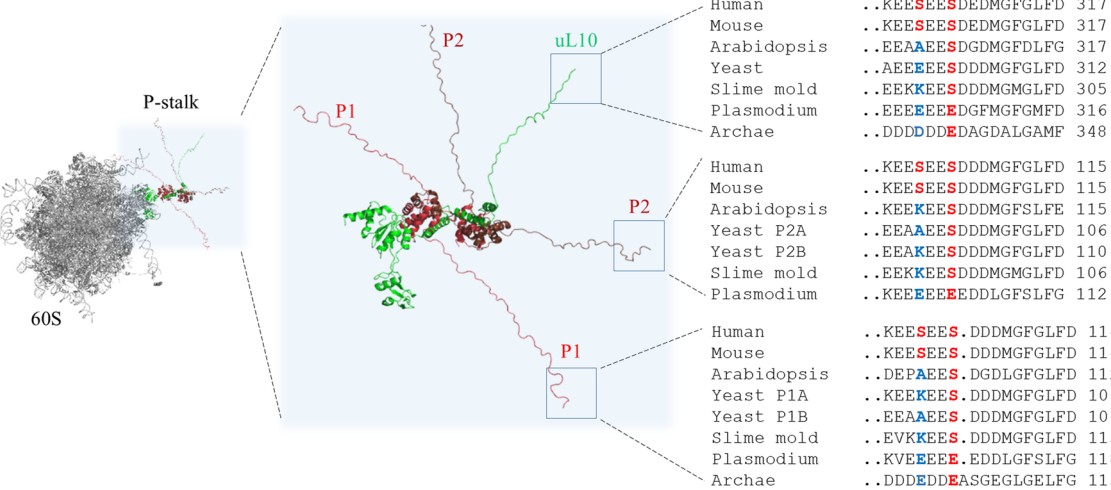

**Figure 1.  Structural representation of the ribosomal P-stalk with the C-terminal amino acid alignment.**

The left panel shows the structural model of the 60S ribosomal subunit from *H. sapiens* (PDB: 4V6X) with the ribosomal P-stalk proteins complex (boxed). The P-stalk proteins are indicated as follows: uL10—green, P1—red, P2—brown, shown as a cartoon representation. Middle panel, zoom-in on the P-stalk pentameric structure; the structure represents compilation of the data for the 60S subunit (PDB: 4V6X for uL10) and P1/P2 dimer (PDB: 4BEH) with the C-terminal motif involved in ligand binding indicated in boxes. The right panel shows amino acid alignments of the C-terminal motifs of the uL10 protein (top), P2 (middle) and P1 (bottom); human - *Homo sapiens* (uL10—accession number #P05388, P1—#P05386, P2—#P05387), mouse - *Mus musculus* (uL10—#P14869, P1—#P47955, P2—#P99027), Arabidopsis, *Arabidopsis thaliana* (uL10—#O04204, P1—#Q8LCW9, P2—#P51407), Yeast, *Saccharomyces cerevisiae* (uL10—#P05317, P1A—#P05318, P1B—#P10622, P2A—#P05319, P2B—#P02400), Slime mold, *Dictyostelium discoideum* (uL10—#P22685, P1—#P22684, P2—#P22683), Plasmodium, *Plasmodium falciparum* (uL10—#Q94660, P1—#Q8IIX0, P2—#O00806), Archaea - *Haloarcula marismortui* (uL10—#P15825, P1—#P15772). The sequences of the P-stalk proteins were retrieved from the UniProt database. The numbers on the right side of the alignment represent total numbers of amino acid residues in the proteins. The amino acid residues marked in red show the position of serine residues that are phosphorylated by the CK2 kinase; in other species one serine can be substituted with alanine, glutamic acid, or lysine, shown in blue.

interaction with Gcn2, which allows timely adaptation to stress conditions. Molecular dynamics simulations demonstrate that the phosphorylation of the P-stalk CTD induces structural transitions, shifting the CTD conformation from a collapsed/globule structure to disordered coils. Consequently, we postulate, that by altering the net charge through CTD phosphorylation within the P-stalk proteins, the CTD gains conformational freedom allowing specific but transient binding to various protein partners.

## Results

### P-stalk proteins are fully phosphorylated in the steady-state conditions in eukaryotic cells

While it is established that P-stalk proteins are modified by CK2, neither the actual proportion of phospho- and dephospho-forms of the P-stalk proteins nor the physiological importance of such modification are known. We determined the actual phosphorylation status of P-stalk proteins in vivo in yeast *S. cerevisiae* and in a mammalian cell line (MEF cells) using the Phos-tag/SDS-PAGE approach. In this approach, the migration of phosphorylated proteins is changed by the Phos-tag reagent, which facilitates the separation and relative quantification of phosphorylated and dephosphorylated protein populations, followed by Western Blotting analysis with specific antibodies against individual P-stalk proteins. To provide a snapshot of the P-stalk protein phospho-state, we first analyzed bulk cell extracts (Fig. 2A). The protocol developed for rapid cell lysis allowed us to avoid exposure

of the protein to phosphatase/kinase activity. As a control, the protein fractions were treated with either the AP-phosphatase (FastAP, which is active in a broad range of buffers) or the CK2 kinase. The P-stalk proteins from the untreated cell extract migrated more slowly than the AP-treated fractions in both the yeast (left panel) and mammalian (right panel) cells, while the CK2 treatment had no influence on the P-stalk protein mobility. This suggests that the P-stalk proteins in the lysate are phosphorylated. To validate this conclusion, we used a yeast strain (SDPM) where all serine residues within the CTDs were mutated to alanine residues, which abolished the phosphorylation of the P-stalk proteins. The P-stalk proteins from the SDPM strain migrated as a single band at the same level as those from the AP-treated cell extracts, confirming that the lower bands corresponded to the unphosphorylated P-stalk proteins (Fig. 2A). In the next step, we analyzed purified ribosomes from the yeast and mammalian cells. The Phos-tag/SDS-PAGE analysis showed that purified yeast ribosomes constituted a mixed population with phosphorylated and dephosphorylated forms of the P-stalk proteins (Fig. 2B, left panel). This may have represented the heterogeneity of the P-stalk proteins in the cells used for ribosome isolation or the result of dephosphorylation during ribosome purification. In contrast, the ribosomal fraction isolated from the MEF cells displayed a single band of the P-stalk proteins corresponding to the phospho-state (Fig. 2B, right panel).

The experimental data for the human stalk were further supported by the meta-analysis of available high throughput mass spectrometry data from the PeptideAtlas database (Desiere et al, 2006). The human 2024-01 build datasets were used, reference

# A

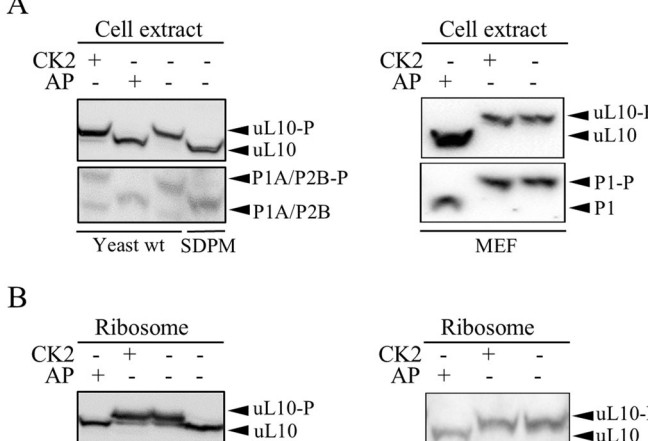

## Figure 2. Quantification of P-stalk phosphorylation in yeast and MEF cell line.

Immunoblot analyses of the total cell extracts (A) or total ribosome fractions (B). The electrophoretic mobility of phosphorylated proteins is changed upon treatments by phos-tag. In addition to the yeast WT strain, the SDPM mutant was used that lacks phosphorylation sites. (A) left panel: line 1, yeast cell extract treated with CK2 kinase (CK2+) lane 2, yeast cell extract treated with phosphatase (AP+); lane 3, untreated cell extract from the WT strain; lane 4, extract from SDPM strain. Right panel: lanes 1, 2, and 3, MEF cell extracts treated with: AP, CK2 and untreated, respectively. (B) Total ribosomal fraction from yeast cells. Lanes 1, 2, 3: ribosomes treated with AP, CK2 and untreated, respectively; line 4, ribosomes from SDPM strain. Right panel: total ribosomes from MEF cell line. Lanes 1, 2 and 3, AP- or CK2-treated and untreated, respectively. The P-stalk proteins were detected using specific antibodies recognizing yeast uL10 and P1A/P2B proteins or human uL10 and P1 proteins. Black arrows show the position of dephospho- and phospho-forms of the P-stalk proteins unmarked and marked with (P), respectively. Source data are available online for this figure.

database: THISP_2024-01-01. Two specific peptides were identified for the human P-stalk proteins: a C-terminal unique peptide KEESEESDDDMGFGLFD shared by P1 and P2 proteins and a unique peptide VEAKEESEESDEDMGFGLFD from uL10. The unbiased MS data showed that P1/P2 and uL10 are mostly double-phosphorylated in the human proteome. Single-site phosphorylation represented about 25% of the population, whereas the dephosphorylated peptide represented only 6% of all phospho-peptide signatures (Fig. EV1, pie-chart, blue part). Similarly, the MS data of the human phosphoproteome suggested that P-stalk proteins exist predominantly (>70%) in the double-phosphorylated state (Fig. EV1, pie-chart, green part). The complementary MS data (PeptideAtlas) for yeast are limited to phospho-proteomic analysis only (YEAST 2022-06 build) (Hollenstein et al, 2021) and do not allow such quantitative estimation.

Having discovered P-stalk proteins in a phosphorylated state within the bulk cytoplasmic ribosomes, our next step was to verify their status in translationally active ribosomes in the polysome fractions. The fractionation of the yeast polysomes showed that the P-stalk proteins were present in the 60S, 80S, and polysome fractions, and the distribution overlapped with the uL23 protein

used as a marker for the 60S subunit. A small fraction of P1/P2 and uL10 proteins was also detected in the ribosome-free fraction (Fig. 3A, left). The analysis of the MEF cells showed a very similar distribution (Fig. 3A, middle panel). We further validated these results using native tissue and examined polysome profiles from mouse liver. As in the yeast and MEF samples, the P-stalk proteins were detected across all polysomal fractions; in this case, no ribosome-free population of the P-stalk protein was detected (Fig. 3A, right panel). The Phos-tag/SDS-PAGE analysis revealed that the P-stalk proteins were mostly present in a phosphorylated form in the yeast, MEF cells, and mouse liver tissue (Fig. 3B). As a final approach to characterize the phospho-state of the P-stalk proteins, the whole cell lysates from the yeast and MEF cells were subjected to AP treatment and, subsequently, the polysomes were resolved by ultracentrifugation. The AP treatment of the isolated cell extracts prior to fractionation did not affect the overall shape of polysomal profiles in the yeast (Fig. 4A, left) or mammalian (Fig. 4A, right) cells, however, the yeast fractions showed a slight decrease in the amount of heavy polysomes. The AP treatment caused a clear shift of P-stalk protein bands toward lower molecular mass, indicating their dephosphorylation by AP (Fig. 4B, lower panel).

Taken together, these data show that, in the steady-state conditions, the P-stalk proteins are predominantly phosphorylated.

## Phosphorylation of P-stalk proteins optimizes the activity of translation factors without a significant effect on the rate of protein synthesis

P-stalk proteins play an important role in the recruitment of trGTPases to the ribosome and contribute to GTPase activation, which prompted us to study the effect of phosphorylation on the binding affinity for trGTPases and GTPase activity using in vitro assays. First, we assessed the interactions of the isolated P-stalk with trGTPase eIF5B as a model system using the MicroScale Thermophoresis (MST) approach. The natively assembled yeast pentameric P-stalk complex was obtained by thrombin cleavage and release of the pentameric complex using an established protocol (Grela et al, 2010) (Fig. EV2A). Two forms of P-stalk complexes were prepared, a native one (phosphorylated) and an AP-treated (dephosphorylated) complex (Fig. EV2B,D). The native-MS and Phos-tag analysis showed that the majority of the P-stalk proteins in the native complex were phosphorylated, however, in the case of uL10, mixed fractions were observed, whereas the P1/P2 proteins were mostly phosphorylated. The AP treatment led to complete dephosphorylation of the proteins (Fig. EV2D). For the MST analysis, the P-stalk complex was titrated with increasing concentrations of eIF5B. The saturation curves were obtained for phosphorylated and dephosphorylated P-stalks, which allowed calculation of the dissociation constant, with $K_d = 4.6 \pm 0.8 \mu M$ and $11.2 \pm 2 \mu M$ for dephospho- and phospho-P-stalk complexes, respectively (Fig. 5A). Surprisingly, the native phosphorylated P-stalk showed two times lower affinity for eIF5B than the dephosphorylated one, but considering the mixed fraction of uL10, the result may be underestimated.

In the next experiment, we examined the effect of the phosphorylation of P-stalk proteins on the rate of trGTPase-dependent GTP hydrolysis. Two types of ribosomes were used in the analysis: the WT with phosphorylated P-stalk proteins and

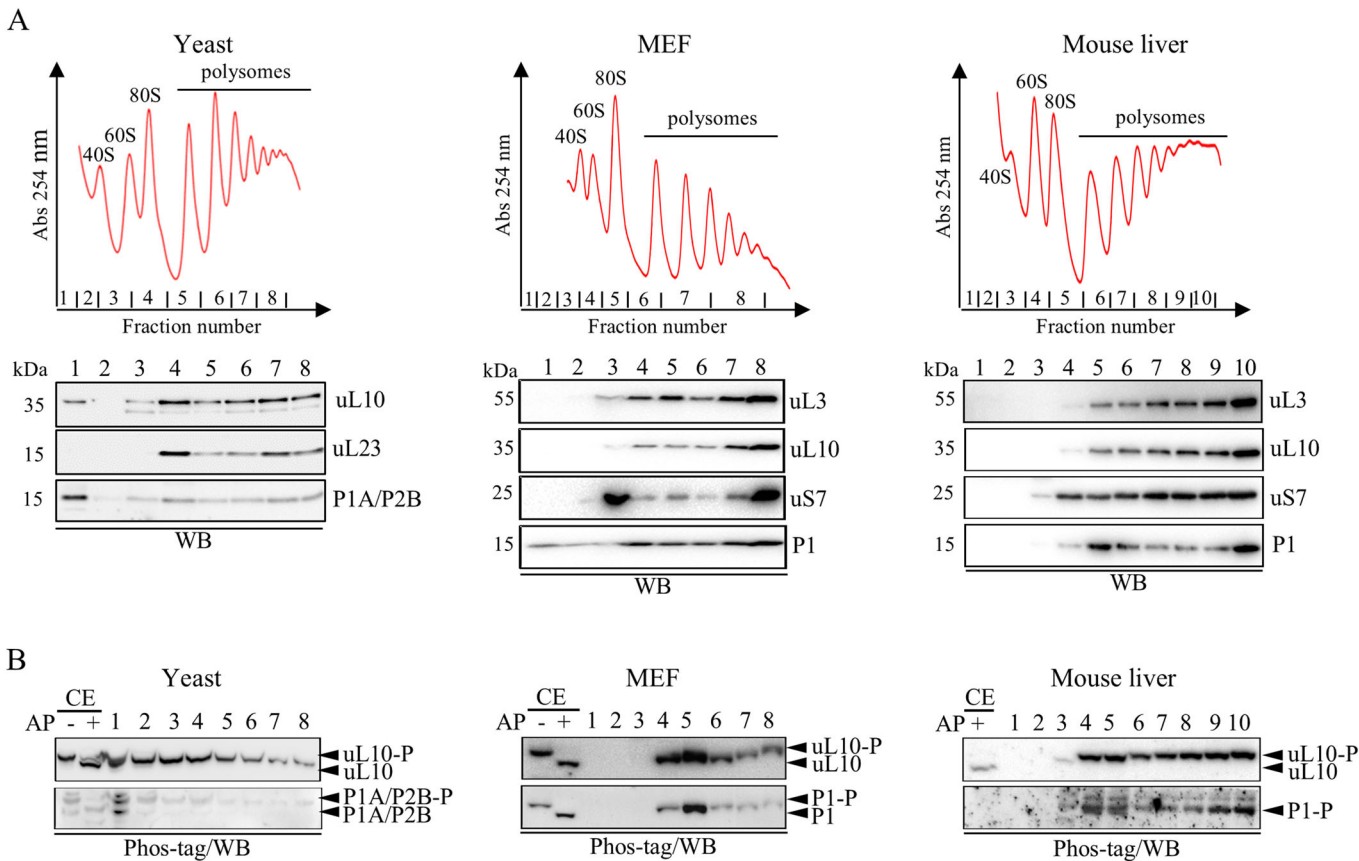

**Figure 3. Phosphorylation of P-stalk proteins in 60S subunits, 80S monosomes and polysomes.**

(A) Polysome profiles obtained from yeast, MEF cells and mouse liver cells (upper panel). The position of 40S, 60S, 80S and polysomes is indicated. The P-proteins and marker ribosomal proteins were visualized by immunoblotting (lower panel). (B) Phos-tag analysis of protein phosphorylation. Alkaline phosphatase treated (+AP) cell extracts (CE) were used to determine position of phosphorylated (−AP, upper band) and dephosphorylated (+AP, lower band) forms of P-proteins. Bands corresponding to both forms are marked with arrows. Source data are available online for this figure.

ribosomes from the SDPM yeast mutant strain with S to A mutations within all the P-stalk proteins. The phospho-status of the P-stalk proteins was confirmed by the Phos-tag/SDS-PAGE analysis, showing homogenous sample preparation, with two forms of ribosomes: phosphorylated isolated from the WT cells and non-phosphorylated from the SDPM mutant strain (Fig. EV2C). In addition, r-protein content was verified by MS analysis, showing that ribosomes from WT and SDPM mutant strains contained the full set of r-proteins. The purified 80S ribosomes were used to stimulate the GTP hydrolysis of various trGTPases, including eIF5B, eEF1A, and eEF2. The stimulatory effect of the ribosomes was observed for all the trGTPases tested, regardless of the P-stalk configuration. To our surprise, the ribosomes with the dephospho-mimicking P-stalk displayed a higher rate of GTP hydrolysis stimulation than the phosphorylated form; in all cases, we observed an approx. 2-fold increase in the level of stimulation, compared to the WT ribosomes (Fig. 5B–D). This can be explained by the higher affinity of the dephosphorylated P-stalk for the trGTPase eIF5B, which is likely to lead to higher GTPase rates at the sub-saturating ribosome concentrations ($k_{cat}/K_M$ conditions) used in these experiments. The higher rate of the GTP hydrolysis in the case of the SDPM mutant compared to the WT ribosomes prompted us to

investigate the effect of P-stalk phosphorylation on the accuracy of decoding, because decoding fidelity is linked to the rate of elongation, and the P-stalk is also involved in maintaining translational fidelity (Wawiorka et al, 2017). The dual-luciferase reporter assay was used to evaluate the decoding error rate in vivo as misincorporation events of near-cognate codons placed at different positions in the reporter gene. The analysis showed increased misreading in the SDPM mutant, showing that the lack of P-stalk phosphorylation reduces the accuracy of decoding (Fig. EV3).

Next, we tested the functional effect of the P-stalk phosphorylation on translation. A fully reconstituted translation system was used to study each step of translation elongation in a codon-resolved manner (Ranjan et al, 2021). As above, two types of ribosomes were used: the WT with phosphorylated P-stalk proteins and the ribosomes from the SDPM yeast mutant strain (Fig. EV2C). First, we tested whether the phosphorylation of the P-stalk proteins affects the first round of translation elongation by monitoring the formation of dipeptides (Met-Phe) in real-time. The rate of Met-Phe formation was comparable in both types of the ribosomes tested (Fig. 6A). These data indicate that the presence or absence of the phosphate group on the P-stalk proteins has no significant

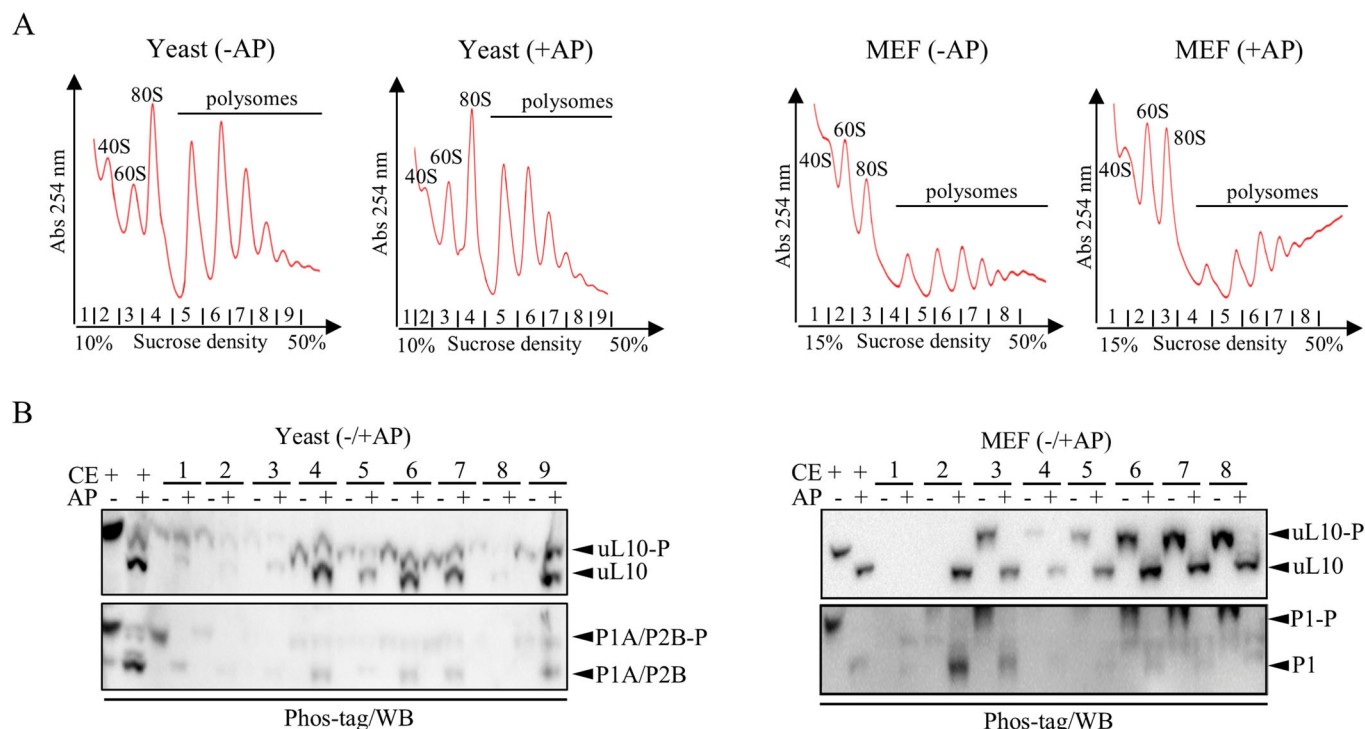

**Figure 4. Analysis of the phosphorylation state of P-stalk proteins bound to ribosomes in yeast and MEF fractions.**

(A) Polysome profiles obtained from yeast and MEF cells upon alkaline phosphatase (AP) treatment. The position of 40S, 60S, 80S and polysomes is indicated. (B) Phos-tag analysis of the P-proteins within each fraction. Corresponding fractions from non-treated (−AP) and treated (+AP) samples were loaded back-to-back for better comparison. The positions of P-proteins are indicated by arrows. Alkaline phosphatase-treated cell extracts (CE; +AP) were used as control to determine the electrophoretic mobility of dephosphorylated proteins. Source data are available online for this figure.

effect on the first round of decoding and peptide bond formation. To probe the influence of the P-stalk phosphorylation on the second round of translation elongation including a translocation event, the same assay was conducted using a preformed 80S complex (with WT or SDPM ribosomes) carrying a dipeptidyl-tRNA (MetPhe-tRNA$^{Phe}$) in the A site and deacylated tRNA$_i^{Met}$ in the P site and a Val-tRNA$^{Val}$ ternary complex reading the next GUU codon (Fig. 6B). The results showed that the rate of tripeptide formation (Met-Phe-Val) for both types of ribosomes was also comparable. These data indicate that the presence or absence of the phosphate group on P-stalk proteins has no significant effect on the analyzed steps of translation elongation, including the activity of eEF2 and eEF3.

Overall, these results indicate that phosphorylation decreases the affinity of the P-stalk complex for trGTPase, leading to slower GTP hydrolysis at sub-saturating ligand concentrations; this, in turn, has a positive effect on decoding accuracy. However, there was no significant effect on the rate of translation elongation, including peptide bond formation and translocation.

## Phosphorylation of stalk proteins optimizes ribosome function

In the next set of experiments, we investigated the effect of the phospho-state of the P-stalk on translation in vivo using the WT and SDPM mutant strains. First, the classical growth rate analysis showed that the absence of P-stalk phosphorylation had a small but

statistically significant negative effect on cell growth (Fig. EV4A). We then analyzed the growth of the cells at the single-cell level, evaluating several aspects of the yeast lifespan, which showed that the SDPM mutant strain had similar parameters to the WT strain, albeit with a reduced reproductive lifespan (Fig. EV4B). The overall translational efficiency in vivo was not significantly affected, as the kinetics of the [$^{35}$S]-methionine incorporation into the newly synthesized polypeptides showed no significant differences between WT and SDPM, supporting the in vitro analysis performed at the single-codon resolution (Fig. 7A). Furthermore, the ribosome half-transit times ($t_{1/2}$), representing the average processivity of the ribosome, were similar for WT and SDPM (Fig. 7B). Also, the quantification of the WT and SDPM polysome profiles showed an indistinguishable polysome-to-monosome ratio (P/M ratio) on both rich (Fig. 7C) and minimal (Appendix Fig. S1) media. Overall, the yeast fitness analyses indicate that the phosphorylation status of the P-stalk has a small but significant effect on cell growth, but not on the overall performance of the translational machinery assessed in vivo.

Next, we compared the translational processivity of the WT and SDPM mutant strains on the genome-wide scale by high-resolution ribosome profiling, HrRP (Wu et al, 2019), at the logarithmic growth phase. The ribosome footprint libraries were prepared using a combination of elongation inhibitors, tigecycline and cyclohex-imide, which allowed distinguishing two distinct ribosome functional states, reflecting the decoding and translocation steps of the elongation cycle. The pre-accommodation state (PreAcc) entails

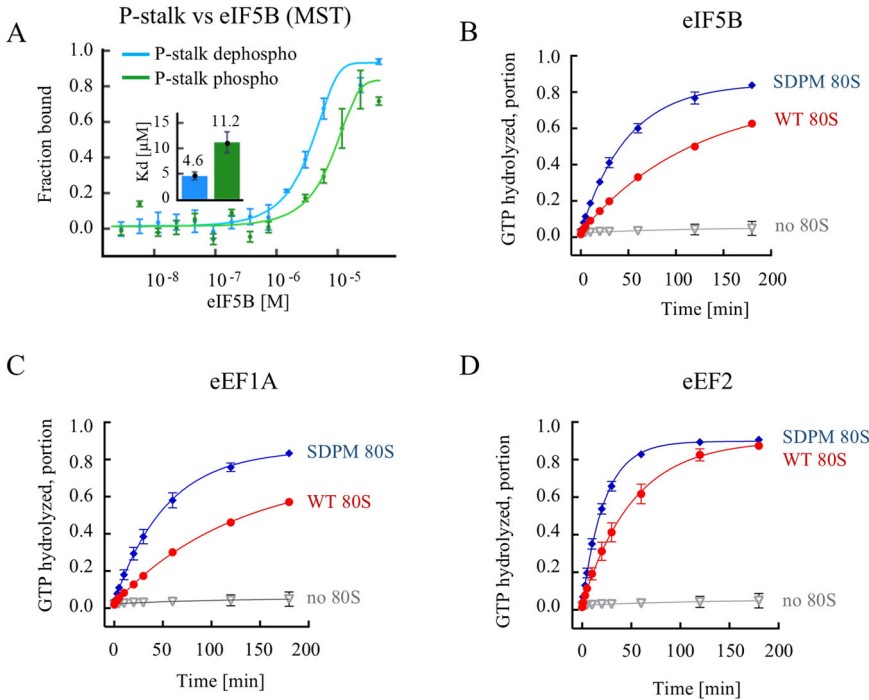

**Figure 5. Effect of phosphorylation on the interaction of P-stalk proteins with trGTPases.**

(A) Binding affinity of eIF5B to phospho P-stalk (green) or dephospho P-stalk (blue) estimated by MST. The P-stalk complex was titrated with the excess of eIF5B. Data presented as mean ± SEM of $n = 3$, technical replicates. Inset, equilibrium dissociation constants ($K_d$): 4.5 ± 0.8 μM, 11 ± 2 μM, respectively. (B–D) GTPase activity of eIF5B (B), eEF1A (C), and eEF2 (D) in the presence of 80S ribosomes from WT (red) or SDPM (blue) strains or in the absence of the ribosomes (gray). Data are presented as mean ± SEM of $n = 3$, biological replicates. The average GTPase rates were 0.008 ± 0.001 min$^{-1}$ with WT and 0.020 ± 0.002 min$^{-1}$ with SDPM ribosomes for eIF5B (B); 0.008 ± 0.001 min$^{-1}$ with WT and 0.019 ± 0.003 min$^{-1}$ with SDPM ribosomes for eEF1A (C); and 0.019 ± 0.003 min$^{-1}$ with WT and 0.044 ± 0.004 min$^{-1}$ with SDPM ribosomes for eEF2 (D). Source data are available online for this figure.

ribosomes with an open A site, which provides information on the decoding process, whereas the pre-translocation state (PreTrans) corresponds to ribosomes with an occupied A site before tRNA translocation is completed. The ribosome-protected footprints of the two states differ in the ribosome-protected footprint size (RPF), i.e., 21 nt RPFs corresponding to PreAcc and 28 nt RPFs corresponding to PreTrans states (Wu et al, 2019). The initial correlation analysis using scatter plots of RNAseq and RIBOseq data showed no substantial changes in the transcriptome and translatome (Fig. EV5A,B). Furthermore, the analysis of the average ribosome occupancy of all genes plotted against CDS, which can be considered a global measure of elongation and termination efficiency, showed no changes either (Fig. EV5C). To characterize the ribosome-specific processivity, we analyzed the distribution of the 21 nt and 28 nt RPFs describing two discrete functional states of elongating ribosomes. The analysis showed a similar bimodal distribution of the 21 and 28 nt RPFs in WT and SDPM, indicating that, in the steady-state conditions, translation elongation in the SDPM mutant did not differ from that in WT (Fig. 8A, left panel), and the relative abundance of the 21 nt and 28 nt RPFs was similar in both strains. Thus, our results indicate that the lack of P-stalk phosphorylation does not cause any significant functional perturbations for elongating ribosomes. Additionally, the codon-specific ribosome occupancies for the 21 nt RPFs were correlated between the WT and SDPM samples (Fig. 8B, left panel); the same behavior was also observed for the 28 nt RPFs (Fig. EV5D, left panels). Once

again, all these data support the idea that the phosphorylation of P-stalk proteins is dispensable for translational machinery performance in optimal conditions.

To gain further insight, we used HrRP to analyze the distribution of two RFPs, 21 and 28 nt, to determine the two functional states of elongating ribosomes following induction of ER stress by tunicamycin (Tun). Tun was used as a general stressor inducing proteotoxic stress, which is considered to occur frequently in cells upon various environmental insults. Tun evokes several types of cellular reactions in the cell, especially ER-stress, inducing the UPR-response (Walter and Ron, 2011), a phenomenon found in natural metabolic transitions in yeast, such as the diauxic shift from the fermentative to mitochondrial respiration phase of growth, also driven by Gcn2 (Patil et al, 2004; Tran et al, 2019). Several time points were used for the analysis because, as we have shown, Tun exerts an immediate effect, with HAC1 mRNA splicing already occurring after 15 min of treatment (Appendix Fig. S2). Changes in the abundance of RPFs in the presence of Tun are regarded as stress markers; in particular, the reduction of the 21 nt RPFs is induced by environmental insults, including osmotic or oxidative stresses (Wu et al, 2019). In contrast to the steady-state conditions, we found reduced levels of the 21 nt RPFs upon Tun-induced stress in the WT cells after 15 min of treatment (Fig. 8A, middle panel). In contrast, the SDPM strain was insensitive to the short Tun treatment, as the ratio between the 21 nt and 28 nt RPFs was similar to that in the untreated sample. After 60 min of the Tun

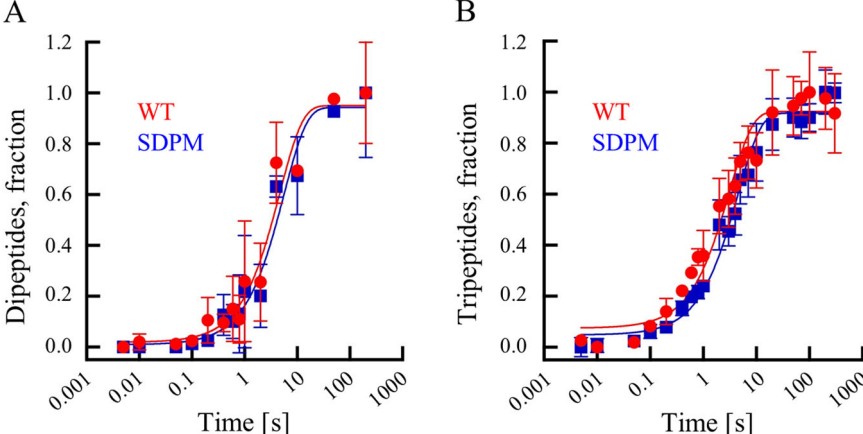

**Figure 6. Translation elongation on ribosomes from the WT and SPDM strains.**

(**A**) Dipeptide (Met-Phe) formation monitored upon rapidly mixing initiation complexes (80S IC) using ribosomes from the WT (red line) and SDPM (blue line) yeast cells, with ternary complexes eEF1A–GTP–[$^{14}$C]Phe-tRNA$^{Phe}$ in a quench-flow apparatus. The extent of peptide formation was analyzed by HPLC and radioactivity counting. Data are normalized to Met-Phe formation with the maximum value in the dataset set to 1 and presented as mean ± SEM of $n = 3$ biological replicates; the average rate is 0.23 ± 0.14 s$^{-1}$ and 0.19 ± 0.10 s$^{-1}$ for WT and SDPM, respectively. (**B**) Tripeptides (Met-Phe-Val) formation monitored upon rapidly mixing 80S ribosomes from WT and SDPM cells carrying MetPhe-tRNA$^{Phe}$ (80S 2 C) with ternary complexes eEF1A–GTP–[$^{14}$C]Phe-tRNA$^{Phe}$. Data are presented as mean ± SEM of $n = 3$, biological replicates; the average rate is 0.34 ± 0.13 s$^{-1}$ and 0.22 ± 0.05 s$^{-1}$ for WT and SDPM, respectively. Source data are available online for this figure.

treatment, the 21 nt to 28 nt RPF ratio was reduced in both the WT and SDPM strains, which indicates that the ribosome response to ER-stress is delayed, but not abolished, by the lack of P-stalk phosphorylation.

Next, to evaluate whether the change in the abundance of the 21 nt RPF is related to specific ribosomal aberrations during elongation, the correlation of codon-specific occupancy for the 21 and 28 nt RPFs was examined. As already shown, the codon-specific occupancy can be a very sensitive measure of ribosomal processivity at the elongation steps, shown as a loss of correlation, with enrichment in several codons, such as proline codons, especially shown for 21 nt RPFs (Wu et al, 2019). In the case of WT and SDPM strains, under control conditions, the codon-specific occupancy showed no enrichment at specific codons (Fig. 8B, left panel). While, after the 15-min Tun treatment, a loss of correlation was recorded, and the occupancies at three codons encoding arginine (CGA, CGG) and proline (CCG) were increased in the WT strain, indicating altered decoding of these codons (Fig. 8B, middle panel) In turn, no effect was observed in the SDPM mutant strain after the 15-min Tun treatment (Fig. 8B, middle panel), but a loss of correlation was observed after 60 min (Fig. 8B, right). After the 60-min treatment, the decoding of the CGA, CGA, and CCG codons was defective in both the WT and SDPM cells. In addition, the analysis of the 28 nt RPFs also revealed an impact on the translocation at the CCG, CGG, and CCC codons (Fig. EV5D), although this effect was more pronounced after 60 min, with a delay in the response in the case of the SDPM strain, as was the case for the 21 nt RPFs.

Taken together, these data show that, although in conditions of normal growth the overall rate of translation is not significantly affected by the phosphorylation of the P-stalk proteins, upon exposure to ER-stress, the ribosome starts to pause at proline and arginine codons, and this response is delayed in SDPM mutant cells. Thus, the phosphorylation of the P-stalk protein plays a role

in stress response by providing a timely translational response to environmental cues.

## Phosphorylation of P-stalk proteins affects Gcn2 activation in ER stress conditions

P-stalk proteins not only recruit trGTPases but also play a role in Gcn2 kinase activation, linking translation to the ISR pathway (Gupta and Hinnebusch, 2023; Inglis et al, 2019; Kim and Zaher, 2022; Masson, 2019). To test the effect of P-stalk phosphorylation on the activation of the ISR pathway, we monitored the phosphorylation status of eIF2α, the target of the Gcn2 kinase, in the WT and SDPM strains upon the Tun treatment. In the WT strain, the eIF2α phosphorylation increased up to 2 h of the Tun treatment and then decreased (Fig. 9A). In contrast, the SDPM strain showed a markedly altered temporal response, with the eIF2α phosphorylation constantly increasing over 8 h (Fig. 9A). In yeast, eIF2α is thought to represent the only substrate of Gcn2 kinase (Dever et al, 1992), but the kinase can be activated via different mechanisms, e.g., by direct interaction with deacylated tRNA that accumulates upon amino acid starvation (Dong et al, 2000; Wek et al, 1995) or by ribosomes carrying uncharged tRNAs in the A site (Sattlegger and Hinnebusch, 2000). Alternatively, Gcn2 can be activated as a result of ribosome stalling independent of the presence of deacylated tRNA (Gupta and Hinnebusch, 2023; Wu et al, 2020). It was suggested, that in yeast cells, Gcn2 is associated with 60S subunits and polysomes (Dong et al, 2000; Ramirez et al, 1991; Zhu and Wek, 1998) and may accumulate on the 60S subunits in stress conditions (Ramirez et al, 1991). To test whether P-stalk phosphorylation modulates this response, we tested the Gcn2 binding to the ribosomes upon the Tun treatment. A 2-h time point was taken for the analysis, where we detected the highest eIF2α phosphorylation in the WT strain. The fractions from untreated (−Tun) and treated (+Tun) samples were loaded side by

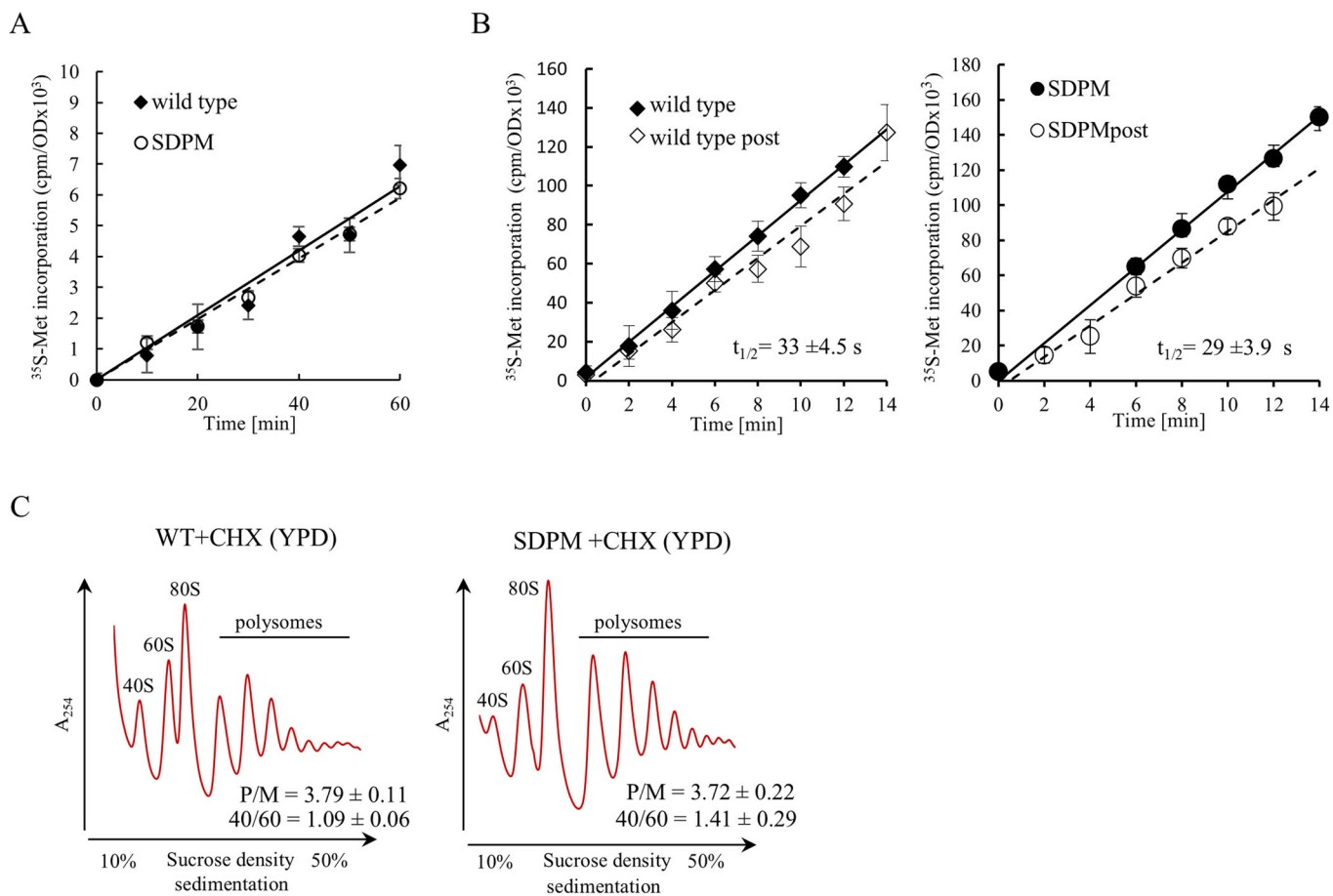

**Figure 7. Translational fitness of WT and SDPM yeast strains in vivo.**

(A) Kinetics of [$^{35}$S]methionine incorporation into newly synthesized polypeptides for wild type (◆) and SDPM (○) cells. Error bars represent ±SD, $n = 3$, technical replicates. (B) Ribosome half-transit times, $t_{1/2}$. The incorporation of [$^{35}$S]methionine into total proteins (nascent and completed) (filled symbols) and completed proteins (open symbols) is shown for each strain. The half-transit time was measured as the displacement between two lines by linear regression analysis and is shown in each panel as an inset. The $t_{1/2}$ values are means ± SD, $n = 3$, technical replicates. (C) Polysome profiles of WT and SDPM strains after CHX treatment. The polysome-to-monosome (P/M) ratio was calculated by dividing the area of the first four polysomal peaks by the area of the peak for the 80S monosome; P/M value is presented as means ± SD, $n = 3$, technical replicates. The positions of individual ribosomal subunits are indicated. Source data are available online for this figure.

side for better comparison. Gcn2 was probed with anti-Gcn2 antibodies (Fig. 9B). The analysis showed that upon the Tun treatment, Gcn2 accumulated on the 60S fraction in the WT strain, whereas this effect was not prominently observed in the SDPM strain (Fig. 9B).

These experiments show that the phosphorylation of P-stalk proteins may play an important role in the timely activation of Gcn2 kinase. The timing of Gcn2 activation is modulated by the possible accumulation of Gcn2 on the 60S subunit, depending on the phosphorylation of the P-stalk, with more rapid activation and adaptation of cells correlating with the phosphorylated P-stalk.

## Structural dynamics of P-stalk protein C-termini is affected by phosphorylation

The CTDs of P-stalk proteins, i.e., the parts proposed to interact with trGTPases, were shown to have unstructured properties in the solution (Lee et al, 2012; Mishra and Hosur, 2015); however, the homologous CTD of the archaeal P1-protein is able to form an α-

helix upon trGTPase binding (Murakami et al, 2018; Nomura et al, 2012; Tanzawa et al, 2018). To understand the effect of phosphorylation on the CTD structure, we performed molecular dynamics simulations (MD) using CTD sequences comprising the C-terminal 22 amino acid residues (Appendix Fig. S3), starting the simulations with a linear chain or an α-helix. We used sequences from human P-stalk proteins, which are doubly phosphorylated, yeast *Saccharomyces cerevisiae*, which have only one phosphorylation site, and an Archaeon *Haloarcula morismortui*, where glutamic acid replaces the invariant serine residues present in eukaryotic sequences (Appendix Fig. S3).

In the CTD peptides, phosphoserine residues are located at positions 9 (human specific) and 12 (conserved in all eukaryotes), framed by lysine and glutamic acid residues (Appendix Fig. S3). The MD analysis revealed that phosphorylation of the serine residues in human P1/P2 and yeast P1 proteins induces a conformational constraint leading to a bent conformation at the N-terminal part of the peptide (Fig. 10A,B, left panel) and (Appendix Figs. S4 and S7), while P1 from Archaeon protein

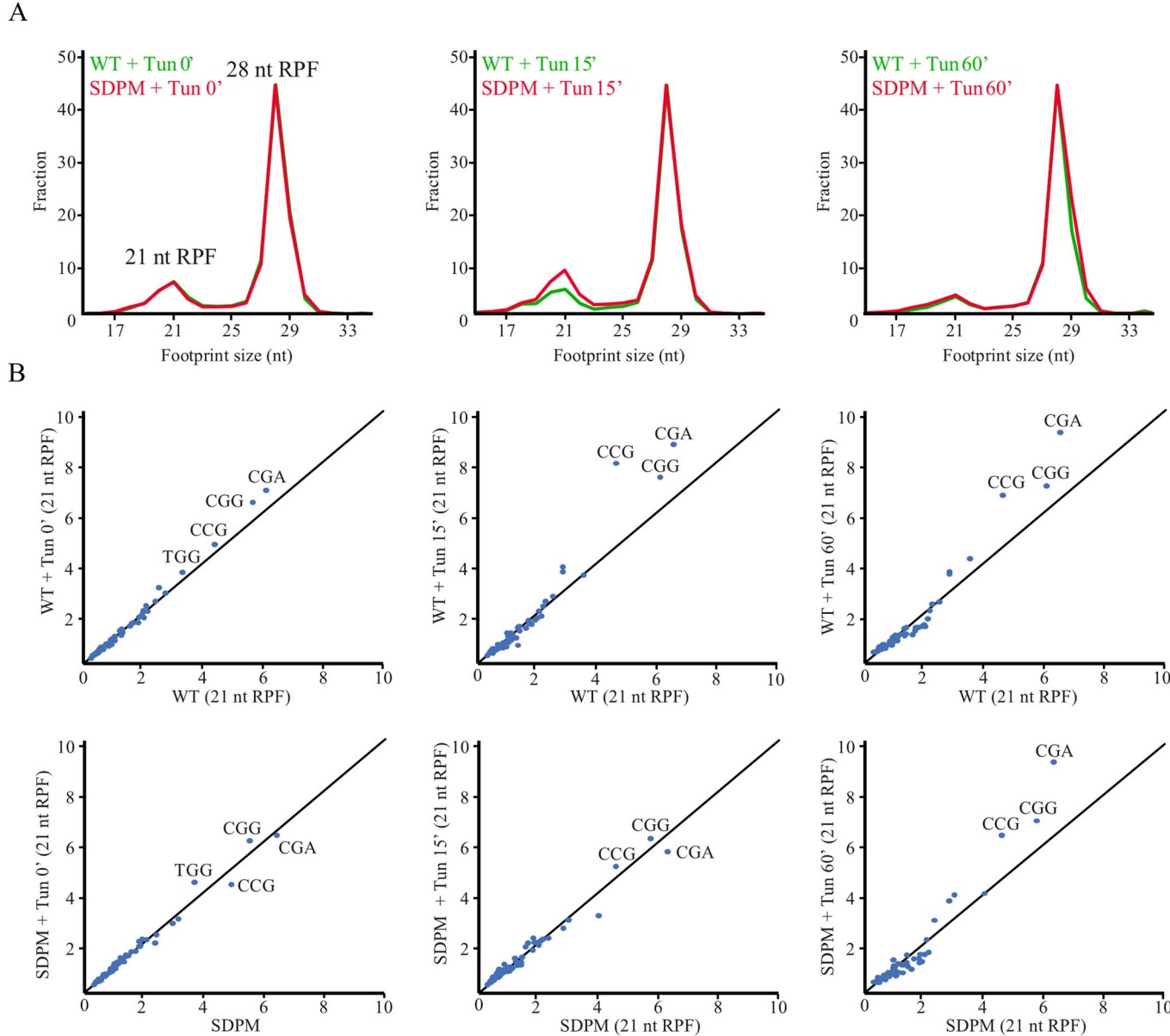

**Figure 8. The relative abundance of 21 nt and 28 nt RPF and specific codon occupancies within 21 nt RPF in steady-state conditions and upon Tun treatment.**

(A) The relative abundance of 21 nt and 28 nt RPF in the course of Tun treatment of WT (green) and SDPM (red). (B) The correlation of specific codon occupancies within 21 nt RPF, corresponding to part A. Specific overrepresented codons CGA, CGG for arginine and CCG for proline are indicated. Source data are available online for this figure.

showed relaxed, extended conformation (Fig. 10C). This bent conformation arises from the network of electrostatic interactions. Specifically, phospho-serine #9 and #12 form salt bridges with positively charged lysine residues, Lys5 and Lys1, respectively, which stabilizes the U-bent shape (Appendix Fig. S4, inset). The conformation is further stabilized by additional salt bridges, e.g., between Lys5 and Glu3 and between Lys1 and Asp14. The formation of stable salt bridges between phosphorylated serine and neighboring positively charged residues restricts the conformational freedom and decreased structural heterogeneity at the N-terminal part of the analyzed eukaryotic peptides. Consequently, the applied structural constraints lead to a structural transition

from the collapsed/globule-to-coil extended structure (Fig. 10A,B, left panels). Analyzing secondary structure transitions over time during the MD simulations, as shown in the bottom panel of Fig. 10A,B, especially for human proteins, shows that the U-bent conformation is evident, as bends (marked in green) and turns (marked in yellow) at the N-terminal part of the peptide, which persist almost throughout the entire simulation. This persistence indicates that the U-bent conformation is highly stable and suggests that the phosphorylated peptide adopts a consistent and reproducible structural motif. This stability and the presence of these bends imply that phosphorylation induces a robust conformational constraint, promoting a uniform structural state that minimizes

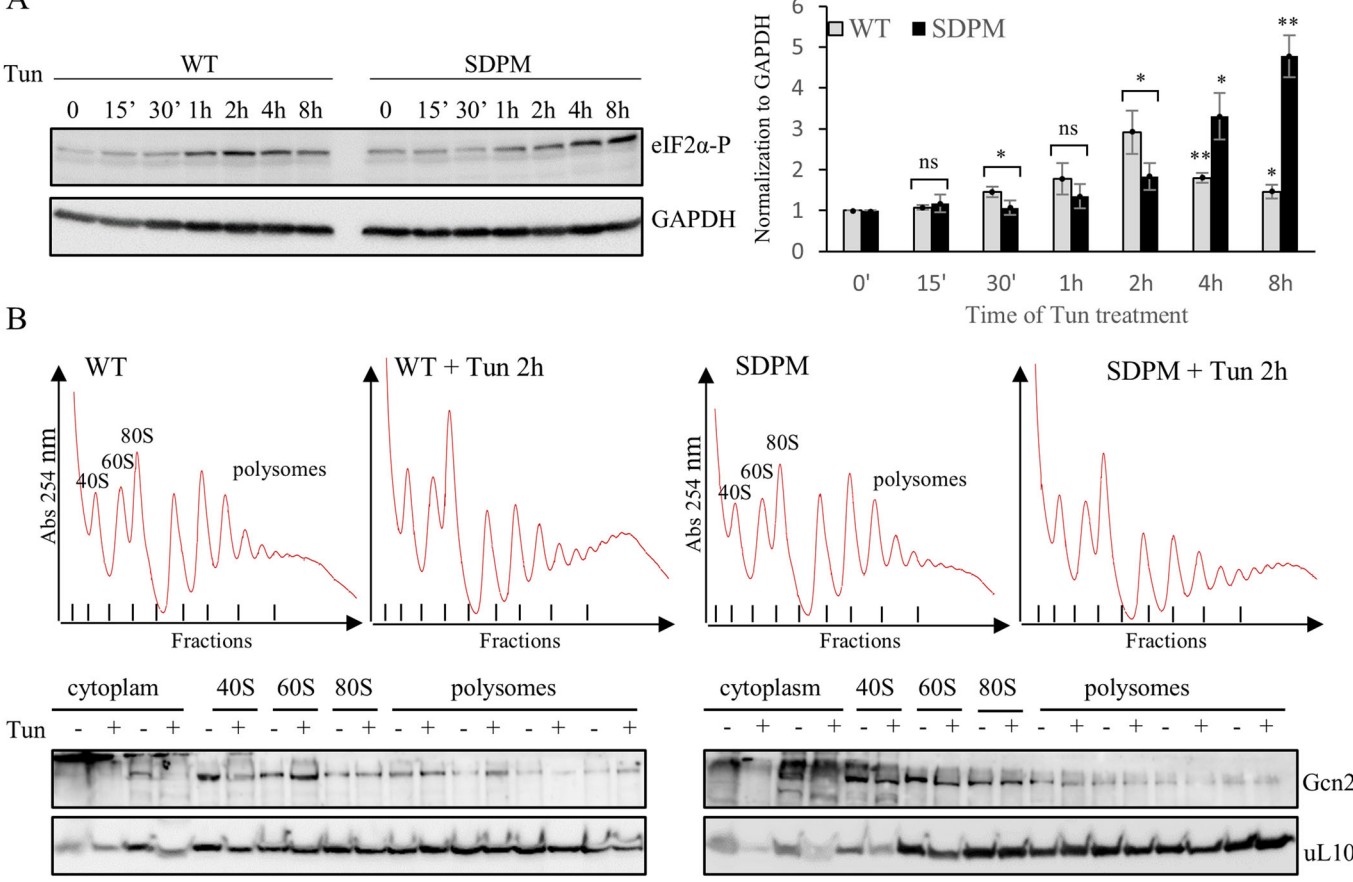

**Figure 9. Ribosome-dependent eIF2α phosphorylation and Gcn2 association with the ribosomes.**

(A) eIF2α phosphorylation after Tun-induced ER-stress probed with specific anti-phospho-eIF2α antibodies; GAPDH was used as loading control. Left panel, time course of phosphorylation. Right panel, quantification of eIF2α phosphorylation relative to GAPDH loading control, normalized to non-treated cells. Error bars represent SD *$p < 0.05$; **$p < 0.001$, determined by Student's $t$-test, $n = 4$, biological replicates. (B) Gcn2 binding to the ribosomes. Top panel, polysome profiles from non-treated ($-$Tun) and Tun-treated ($+$Tun) (2.5 µg/ml of tunicamycin for 2 h) WT and SDPM yeast cells. Lower panel, Western blot analysis with antibodies against Gcn2 kinase and uL10 ribosomal protein. Fractions from control and Tun exposed cells are marked with $-/+$. Source data are available online for this figure.

conformational variability. This structure corresponds to a new global minimum, leading to a significantly less rugged conformational landscape and a narrower and simpler peptide free energy landscape (FEL) (Fig. 10A,B, right panel). This stabilizing effect and structural transition is also observed in the peptides of human P2 and uL10 (Appendix Figs. S4 and 5), having similar changes in FEL minima (Appendix Fig. S6).

Despite very conserved features, the sequences of the P-stalk proteins display some differences in human and yeast (Fig. 1; Appendix Fig. S3). In yeast, the P1/P2 proteins diverge into four distinct elements, forming two hetero-dimers, P1A-P2B and P1B-P2A (Tchorzewski et al, 2000a; Tchorzewski et al, 2003). Contrary to the human counterparts, each of the P-stalk proteins contains only one phosphorylated serine residue and only one closely placed Lys residue in the P1A-P2B dimer, while P1B-P2A has only a distal Lys and uL10 has none at all (Appendix Fig. S3). As a result, the U-bent occurs only sporadically and is not as stable as the human CTD. Notably, the presence of a Lys residue in the vicinity of the phosphorylated serine makes the U-bent conformation more stable during simulations conducted for P1A and P2B than for other

peptides (Fig. 10B; Appendix Figs. S7 and 8). The peptides from the P1B and P2A proteins do not display the fixed U-shape bent but can adopt a variety of conformations (Appendix Fig. S7), resulting in a more even FEL distribution (Appendix Fig. S9). In Archaea, there are only two different P-stalk proteins, uL10 and P1, which form a homodimer attached to the uL10. P1 and uL10 share similarity with the CTDs of eukaryotic P-stalk proteins but are not phosphorylated (Fig. 1; Appendix Fig. S3). The MD simulations revealed that, due to the absence of phosphorylated serine residues, there was no discernible tendency towards forming a U-bent conformation (Fig. 10C; Appendix Figs. S10 and 11) with an even FEL distribution (Appendix Fig. S12).

The MD simulations were also conducted on all types of CTDs with peptides initially formed as α-helices. Although all the peptides ultimately transitioned to an unstructured state, those from Archaea retained their helical conformation significantly longer than the CTDs from the other species (Appendix Figs. S13, 14 and 15). This enhanced stability of the archaeal α-helices is consistent with a reported structural analysis, which indicates that archaeal peptides can form an α-helix when interacting with

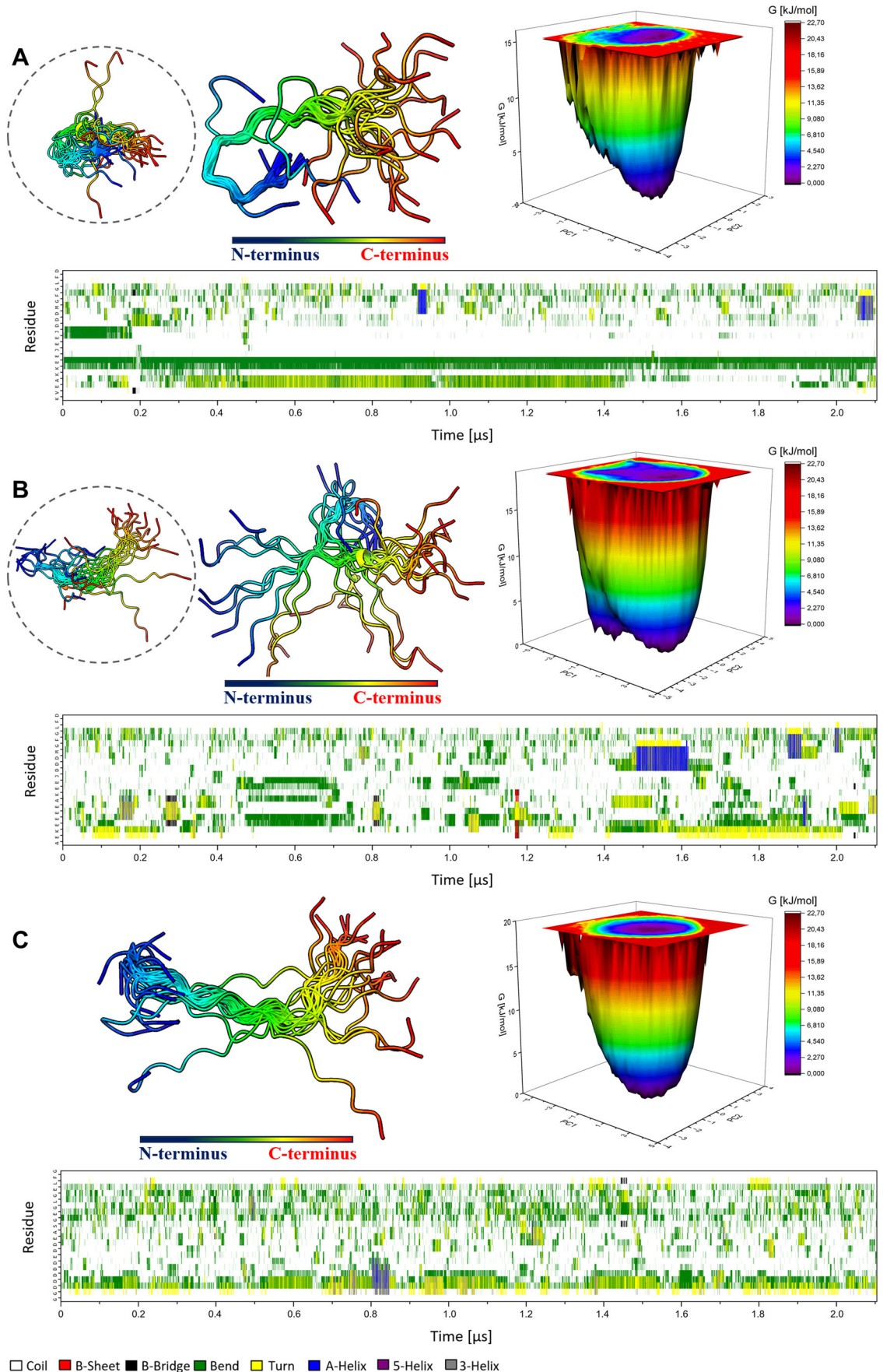

Figure 10. MD simulation of structures of the unphosphorylated and phosphorylated forms of P-stalk C-terminal peptides.

(A–C) Representative structures of the most populated clusters from the MD simulation of the human, yeast and archaeal P1 proteins, respectively. Insets in (A) and (B) display the representative structures of the most populated clusters from the MD simulation of the unphosphorylated P1-proteins, respectively. The right panels in (A), (B), and (C) present free energy landscape plots of C-termini peptides folding in unphosphorylated and phosphorylated state. The bottom panels in (A), (B), and (C) depict C-termini peptide secondary structure transitions over time during the MD simulations.

trGTPases (Murakami et al, 2018; Nomura et al, 2012; Tanzawa et al, 2018). It seems that archaeal CTDs tend to maintain α-helices, while the phosphorylation of eukaryotic CTDs destabilizes α-helix formation, resulting in a U-bent conformation with the very C-termini of the CTD adopting a coil-like conformation. To address the role of lysine residues, the MD simulations were performed with lysine-to-alanine mutant forms of the C-terminal P-stalk peptides (Appendix Fig. S16). The analysis showed, that the absence of lysine made the peptides even less likely to form the bends and turns responsible for the U-bend structure in the N-terminal part of the peptide, suggesting that lysine residues play a crucial role in stabilizing the U-bend conformation.

In conclusion, the MD simulations demonstrate that the presence of phospho-serine sidechains and Lys residues in their vicinity stabilizes the U-bent structure in the CTD and stimulates the collapsed/globule-to-coil transition, an effect particularly evident in the case of human P-stalk proteins. The simulations demonstrate the remarkable evolutionary transition of CTDs. In Archaea, which lack CTD phosphorylation, the coil-like structure is readily maintained and may also form a stable α-helix. In yeast, the CTDs exhibit transitional properties; the CTDs from the dimer (P1B-P2A) are similar to the archaeal type, while the others (P1A-P2B) resemble properties of human CTDs. In human, the double phosphorylation of the CTDs is crucial for stabilizing the U-bent structure, resulting in an extended coil conformation.

## Discussion

The ribosomal P-stalk located on the 60S subunit is primarily responsible for interactions with trGTPases and facilitate GTP hydrolysis. However, the eukaryotic ribosomal P-stalk has additional functions beyond its canonical role in translation. It also provides an interacting platform for several additional auxiliary factors, such as Ribosome Inactivating Proteins, RIPs (Grela et al, 2017; Grela et al, 2019) and also interacts with the Gcn2 kinase, which plays an important role in cellular response to adverse growth conditions (Masson, 2019) and can be perceived as a key link between the Integrated Stress Response signal transduction pathways and the ribosome (Wu et al, 2020). Consequently, the molecular characteristics of the P-stalk may be tailored to interact with a wide range of diverse partners.

Thus, the question arises: how did the eukaryotic P-stalk achieve such unique, pleiotropic, and adaptive characteristics to interact with a variety of protein partners? Considering the high degree of flexibility of CTDs from the P-stalk proteins, which have IDP properties, these unique CTD architecture may allow promiscuous interactions with different partners on various occasions, following the so-called "fly-casting" mechanism (Shoemaker et al, 2000). This phenomenon was already proposed for the bacterial L7/12 stalk, which is able to sample a large volume around the ribosome, being

critical for functional interactions with trGTPases (Bernado et al, 2010; Diaconu et al, 2005). The key aspect of the IDP is its ability to interact with a broad range of partners with high specificity and modest affinity (van der Lee et al, 2014), and these interactions are often achieved or regulated by phosphorylation. P-stalk proteins belong to the group of ribosomal proteins that undergo phosphorylation (Hasler et al, 1991; Zambrano et al, 1997; Zinker and Warner, 1976), but the phosphorylation state and the role of this post-translational modification has not been revealed.

Here, we demonstrate that, in optimal growth conditions, P-stalk proteins are found exclusively in a phosphorylated state, and this feature is conserved in Eukaryotes (Fig. 1). Considering the functional implications of phosphorylation, such modifications are typically transient (Nussinov et al, 2012), and several studies have indicated that the phosphorylation levels of P-stalk proteins may vary (Maniratanachote et al, 2006; Pan et al, 2014; Pan et al, 2013; Zampieri et al, 2001). However, despite examining a wide range of stress conditions, we found that these proteins remained phosphorylated, regardless of the stress conditions tested. This observation is supported also by recent studies examining the effect of UV on cellular metabolism (Sinha et al, 2024), where P-stalk proteins were found phosphorylated.

Our earlier study suggested that phosphorylation decreases the affinity of the P-stalk proteins toward RIPs (Horbowicz-Drozdzal et al, 2021). Importantly, here we show that the phosphorylation of the P-stalk also decreases its affinity toward the trGTPase, eIF5B. However, this information is in contrast to the previous report, which showed that phosphorylation of the P2 protein can promote interaction with eEF2 (Vard et al, 1997). In this case, the recombinant P2 protein was used as the analyte, which may have biased the results, as P2 protein is known to tend to form high-molecular mass oligomers without its natural P1 partner (Grela et al, 2008b; Tchorzewski et al, 2000a; Tchorzewski et al, 2000b). The observed lower affinity has a biological impact. We observed that it reduces the stimulatory effect on GTP hydrolysis by trGTPases. Considering the ribosome performance, the speed and accuracy of decoding are fundamental elements ensuring correct metabolic fitness of living cells. The GTP hydrolysis rate is crucial for optimizing both the speed and accuracy of ribosome action and represents a balance between these two fundamental parameters (Wohlgemuth et al, 2010). However, the measured rate of translation in a codon-resolved manner, showed that the overall speed of translation is not regulated by the CTD phosphorylation. However, as we have already reported, the P-stalk also contributes to the accuracy of decoding (Wawiorka et al, 2017), and our finding that the SDPM mutant has a higher error rate due to the higher GTP hydrolysis rate further supports this idea. This finding is supported by reports showing that, in contrast to the bacterial translation, which seems to be optimized towards a high speed of translation at the cost of fidelity (Wohlgemuth et al, 2010), the ribosomes in Eukaryotes, are slower than in bacteria. The timing of

the reactions occurring during the decoding has a significant role in adjusting the decoding accuracy, which provides superior decoding fidelity in eukaryotes (Holm et al, 2023). Thus, it can be concluded that the phosphorylation of P-stalk proteins optimizes the decoding, probably by controlling the timing of interactions with the ternary complex.

At this point, another question arises: what is the biological outcome of the P-stalk protein phosphorylation beyond translation? We have shown that P-stalk proteins are phosphorylated regardless of the metabolic conditions. While the overall translation rate is not changed, considering phospho- or dephospho-states, the functional effect of the P-stalk phosphorylation is more pronounced upon ER-stress, as shown by the genome-wide analysis. Upon stress, we have shown that ribosomes start to pause specifically at CGA arginine codons, which is in line with a previous report showing that the decoding of the CGA codons may be challenging due to the low abundance of tRNA$^{Arg}$ and its inosine wobble pairing interaction (Letzring et al, 2013). Also, as it was also already reported, successive CGA codons in yeast trigger ribosome stalling, likely through steric effects at both decoding and peptidyl transferase centers (Matsuo et al, 2020), which may lead to the ribosome stalling-dependent activation of Gcn2 kinase (Kim and Zaher, 2022). Indeed, the presented ribosome profiling data were supported by the evaluation of eIF2α phosphorylation upon the Tun treatment. In the WT cells, the eIF2α phosphorylation increased at the onset of stress but decreased over time due to adaptation. In contrast, the SDPM mutant exhibited a delayed response and a constant increase in the eIF2α phosphorylation, indicating the lack of proper adaptation. Interestingly, the effect on eIF2α phosphorylation shows a significant delay in relation to the onset of stress. Although the stress response begins within 15 min, the effect on eIF2α phosphorylation is not observed until after one hour. This suggests that signaling from ribosomes to downstream effectors is buffered, possibly acting like a rheostat, as suggested by recent studies (Sinha et al, 2024). In addition, it has been reported that Gcn2 in yeast may associated with the 60S subunit, especially when cells are exposed to stress conditions (Dong et al, 2000; Ramirez et al, 1991; Zhu and Wek, 1998). Our findings indicate that Gcn2 has a potential ability to bind to the ribosomal 60S subunit, which is enhanced by the Tun-induced stress, and that phosphorylation of the P-protein further strengthens this binding, suggesting that P-stalk phosphorylation may have a beneficial impact on the interaction with Gcn2 and consequently, on the proper development of the stress response.

Another intriguing functional aspect of P-stalk proteins is their ability to interact with a variety of partners. Structural work suggests that the CTD of P-stalk proteins can adopt different structural configurations upon interaction with different protein partners, such as the trGTPases eIF5B (Murakami et al, 2018), eEF1A (Nomura et al, 2012), eEF2 (Tanzawa et al, 2018), and ABCE1 (Imai et al, 2018) and RIPs, such as ricin(Fan et al, 2016), trichosanthin (Too et al, 2009), and Shiga toxin (Kulczyk et al, 2023). Therefore, how does the CTD achieve such adaptability to different partners? An NMR study of human P-stalk proteins has demonstrated that the CTD is unstructured (Lee et al, 2012) and behaves as the IDR in the solution (Mishra and Hosur, 2015). On the other hand, the crystallographic structural studies have shown that the CTD can adopt an α-helical structure when binding to

trGTPases or adopt various extended conformations when binding to RIPs. Our MD simulations suggest that such structural adaptability can be achieved through phosphorylation of the CTD. As it has already been shown, several functional modules within the IDR can be defined as a short linear motif (SLiM); the first one is usually involved in direct interactions with partners, and the other is engaged in the regulation of the interplay with partner protein (Davey et al, 2012). The C-terminal element of human P-stalk proteins (…**KVEAKKEESEESDDD**MGFGLFD) entails two characteristic elements: a highly charged cluster of amino acid residues including phosphorylated serine residues (marked in red) and a hydrophobic part (marked in blue), which represents the main interaction element for various protein partners. The latter sequence, MGFGLFD, can be considered a SLiM #1 involved in the binding of protein partners and adopting various structural organizations upon binding, ranging from linear extended structures (Fan et al, 2016; Too et al, 2009) (as reported in the case of RIPs) to an α-helical fold (Kulczyk et al, 2023; Tanzawa et al, 2018) (characteristic for trGTPases or RIPs). The SLiM #2 (in red) within the CTD is highly charged and contains phosphorylated serine residues. The structure of this peptide has not been solved experimentally, probably due to the high intrinsic flexibility of this region. Based on the MD simulation, we showed that phosphorylation changes the structure of SLiM #2 by stabilizing the U-bent conformation, which provides structural freedom for SLiM #1. The unphosphorylated CTD displays a continuum of conformational ensembles transiently sampling coil-like states and more compact collapsed globules, with a broad range of secondary structures. Thus, the presence of phosphate group, especially in higher Eukaryotes, induces the U-bent shape within the SLiM #2, which provides conformational freedom to the very C-terminal patch of amino acid residues within the SLiM #1. We postulated, that this transition may increase the propensity to interact with a broad repertoire of partners and, on the other hand, destabilize the interactions, providing short-lived transient binding. As demonstrated for other IDPs, particularly those possessing polyelectrolyte properties with numerous Glu and Asp residues, the transition from collapsed/globule to expanded unstructured coils is sharp, indicating that minor changes in the net charge, such as those resulting from phosphorylation, can trigger globule-to-coil transitions (van der Lee et al, 2014). Therefore, it can be postulated that the phosphorylated CTD of P-stalk proteins, with the coil-like conformation, gains conformational freedom, which in turn may allow robust and promiscuous but transient binding to the various protein partners. The proposed CTD structural behavior may be supported by the fact that binding of P-stalk protein CTDs to trGTPases and RIPs is taking place on the various structurally distinct concave extended grooves on the surface of partner proteins (Fan et al, 2016; Imai et al, 2018; Kulczyk et al, 2023; Murakami et al, 2018; Nomura et al, 2012; Tanzawa et al, 2018; Too et al, 2009), acting as an 'entropic trap', reducing the overall free energy of binding.

Taken together, our findings reveal that P-stalk proteins exist exclusively in a phosphorylated state on the ribosomal particles, and the presence of phosphate groups leads to collapsed/globule-to-coil conformational transitions within the intrinsically disordered region of the CTD. This structural shift yields functional outcomes by optimization of ribosome interactions with auxiliary factors, such as trGTPases; however, this post-translational modification

does not affect the overall rate of translation, apart from decoding fidelity. On the other hand, ribosomal particles with phosphorylated P-stalk proteins effectively activate the Gcn2 kinase, contributing to the correct timing of kinase activation. Therefore, we propose that the CTD phosphorylation not only optimizes the ribosome performance in translation per se but has also triggered the expansion of the ribosomal interactome beyond the trGTPases, allowing timely interaction with the Gcn2 kinase. Thus, P-stalk proteins with the IDP region enable eukaryotic ribosomes to serve as an adaptable regulatory hub linking translation with regulatory pathways.

# Methods

### Reagents and tools table

| Reagent/Resource | Reference or Source | Identifier or Catalog Number |
|---|---|---|
| **Experimental Models** | | |
| Yeast strain - BY4741/2 (MATa/alpha his3Δ1 leu2Δ0 met15Δ0 ura3Δ0) | EUROSCARF | Y00000/Y10000 |
| Yeast strain - uL10TH199 (MATa; RPP0-TH199; his3Δ1; leu2Δ0; lys2Δ0; and ura3Δ0) | Grela et al, 2010 | Not applicable |
| Yeast strain - SDPM (BY4741 with mutations S-to-A within all P-stalk proteins at the C-termini, uL10, P1A, P1B, P2A and P2B) | This study | Not applicable |
| Cell line - MEF (DR-Wildtype) | ATCC | CRL-2977 |
| Cell line - NIH 3T3 | ECACC | 93061524 |
| Male albino Swiss mice | Laboratory Animals Breeding, Ilkowice, Poland | Not applicable |
| **Recombinant DNA** | | |
| pDB688 | Provided by Bedwell laboratory | Not applicable |
| pDB868 CGC$_{245}$ | Provided by Bedwell laboratory | Not applicable |
| pJD375 | Provided by Dinman laboratory | Not applicable |
| pJD643 AGC$_{218}$ | Provided by Dinman laboratory | Not applicable |
| **Antibodies** | | |
| Mouse anti-tubulin (DM1A) | Abcam | Ab7291 |
| Rabbit anti-RPLP0 | Abcam | Ab192866 |
| Mouse anti-eIF2α (D-3) | Santa Cruz Biotechnology | Sc-133132 |
| Rabbit anti-RPS5 | Santa Cruz Biotechnology | Sc-100832 |
| Rabbit anti-RPL3 | Proteintech | 11005-1-AP |
| Rabbit anti-RPLP1 | Atlas Antibody | HPA003368 |
| Rabbit anti-P1A/P2B | In house, Boguszewska et al, 2002 | Not applicable |
| Rabbit anti-RPL25/uL23 | Provided by Ed Hurt Laboratory | Not applicable |
| Rabbit anti-GAPDH | Proteintech | 60004-1-Ig |

| Reagent/Resource | Reference or Source | Identifier or Catalog Number |
|---|---|---|
| Rabbit anti-EIF2S1 (phospho S51) | Abcam | Ab32157 |
| Goat anti-Rabbit IgG (H + L)-HRP Conjugate | Bio-Rad | 1706515 |
| Goat anti-Mouse IgG (H + L)-HRP Conjugate | Bio-Rad | 1706516 |
| Rabbit anti-Gcn2 | Provided by Alan Hinnebusch lab | Not applicable |
| **Chemicals, Enzymes and other reagents** | | |
| YPD medium | MP Biomedicals | 114001032 |
| Dulbecco's Modified Eagle Medium- high glucose | Merck | D6429 |
| Fetal bovine serum | Merck | F0804 |
| Penicillin/Streptomycin | Merck | P4333 |
| Minimal Eagle's Medium | Merck | M2279 |
| GlutaMAX | Gibco | 35050-061 |
| Complete Mini EDTA-free Protease Inhibitor Cocktail | Roche | 04693159001 |
| Complete Mini EDTA-free Protease Inhibitor Cocktail | Roche | 5056489001 |
| Bio-Rad Protein Assay Dye Reagent Concentrate | Bio-Rad | 5000006 |
| Bovine Serum Albumin | Merck | A4503 |
| DNase I | Jena Bioscience | EN-173S |
| Puromycin | Serva | 33835 |
| Cycloheximide | Merck | C7698 |
| RNase inhibitor | Merck | R7397 |
| Dulbecco's Buffered Saline | Gibco | 14040133 |
| FastAP Thermosensitive Alkaline Phosphatase | Thermo Fisher Scientific | EF0652 |
| Phos-tag | Wako Chemicals | AAL-107 |
| Clarity™ Western ECL | Bio-Rad | 170-5061 |
| Thrombin | Merck | T4648-1KU |
| **Software** | | |
| Microcal Origin 6.0 | OriginLab Corporation (https://www.originlab.com) | Not applicable |
| ImageLab | Bio-Rad | Not applicable |
| RStudio environment version 4.2.1 | Posit (https://posit.co) | Not applicable |
| NanoTemper NT version 2.2.4 | NanoTemper Technologies | Not applicable |
| UCSF ChimeraX tool | University of California (https://www.cgl.ucsf.edu/chimerax/) | Not applicable |
| **Other** | | |
| Glass beads | Merck | G8772 |
| Corundum | Merck | 8334 |
| Centrifuge MPW-150R | MPW Med Instruments | Not applicable |

| Reagent/Resource | Reference or Source | Identifier or Catalog Number |
|---|---|---|
| Optima™ MAX-TL | Beckman-Coulter | Not applicable |
| Rotor MLA-80 | Beckman-Coulter | Not applicable |
| Optima™ L-100XP | Beckman-Coulter | Not applicable |
| Rotor SW32 Ti | Beckman-Coulter | Not applicable |
| Gradient master 108 | Biocomp | Not applicable |
| Gradient fractionator | ISCO Brendel | Not applicable |
| PVDF membrane | Millipore | IPVH00010 |
| Mini Trans-Blot Cell Module | Bio-Rad | 1703930 |
| ChemiDoc™ MP analyzer | Bio-Rad | Not applicable |
| Monolith NT.115 | NanoTemper | Not applicable |
| Nikon Eclipse E200 | Nikon | Not applicable |
| Monolith His-Tag Labeling Kit RED-tris-NTA 2nd Generation Kit | NanoTemper Technologies | Not applicable |

## Yeast strains

Yeast haploid strains BY4741/2 (*MATa/alpha his3Δ1 leu2Δ0 met15Δ0 ura3Δ0*) purchased from EUROSCARF (Scientific Research and Development, Germany) were used as wild-type strains. The SDPM mutant strain with serine to alanine substitutions within the C-termini of all stalk proteins: uL10, P1A, P1B, P2A, and P2B was derived from the BY4741/2 strains. This mutant was constructed in a stepwise fashion with the use of a pop-in/pop-out strategy (Boeke et al, 1987) using the BY4741 and BY4742 parent strains and URA3-based plasmids containing the mutation within synthetic DNA (Genscript Inc). After confirmation of three different substitutions in each strain, two strains were mated, sporulated, and subjected to tetrad analysis. The resulting spores were genotyped by PCR-seq to identify the quintuple SDPM mutant. The yeast uL10 mutant strain uL10TH199 (MATa; RPP0-TH199; his3Δ1; leu2Δ0; lys2Δ0; and ura3Δ0) used for purification of P-stalk complexes was described previously (Grela et al, 2010). The yeast cells were cultivated in YPD medium (1% yeast extract, 2% peptone, 2% dextrose) at 30 °C.

## Cell lines

MEF cells were purchased from ATCC and were grown in Dulbecco's modified Eagle medium - high glucose (DMEM-Hi) (Merck, cat#D6429) supplemented with 10% FBS (Merck, cat#F0804) and with antibiotics penicillin/streptomycin (Merck, cat#P4333). The NIH3T3 cell line was purchased from ECACC and was grown in Minimal Eagle's Medium (Merck, cat#M2279)

supplemented with 10% FBS, penicillin/streptomycin, and 1X GlutaMAX solution (Gibco, cat#35050-061).

## Yeast extracts and ribosome purification

To obtain cell extracts, the yeast cells were grown to $OD_{600}$ of 0.4–0.6 and harvested by centrifugation at $1000 \times g$ for 3 min at room temperature (RT). Pelleted cells were suspended in RIPA lysis buffer (50 mM Tris-HCl pH 7.5, 150 mM NaCl, 0.1% SDS, 1% NP-40) supplemented with 1x complete Mini EDTA-free Protease Inhibitor Cocktail (Merck, Roche cat#04693159001) and disrupted by vigorous shaking with glass beads (Merck, Sigma-Aldrich #catG8772) followed by centrifugation at $15,000 \times g$ for 20 min at RT. The protein concentration in the supernatant was determined by the Bradford procedure using Bio-Rad Protein Assay Dye Reagent Concentrate (Bio-Rad, cat#5000006) and bovine serum albumin (BSA, Merck, Sigma-Aldrich cat#A4503) used for preparation of the standard curve.

For purification of ribosomes, the yeast cells were harvested by centrifugation in the logarithmic phase of growth when $OD_{600}$ reached 0.5. Then, the cells were washed with ice-cold buffer A (50 mM Tris-HCl pH 7.5, 80 mM KCl, 12.5 mM $MgCl_2$, 1 mg/ml heparin, 2 mM DTT, 1 mM PMSF, 0.5 µM puromycin, 1x complete Mini EDTA-free Protease Inhibitor Cocktail), frozen in liquid nitrogen, and preserved at −70 °C. The following day, the frozen cells were ground with corundum (Merck, Sigma-Aldrich cat#8334) in buffer A supplemented with DNase I (Jena Bioscience, cat#EN-173S). The obtained cell lysate was centrifuged at $105,000 \times g$ (MPW-150R, MPW Med Instruments) for 30 min at 4 °C. For ribosome purification, the clarified lysate was loaded on a 30% sucrose cushion in buffer containing 50 mM Tris-HCl pH 7.5, 500 mM KCl, 12.5 mM $MgCl_2$, 1 mg/ml heparin, 2 mM DTT, and 1 mM PMSF and centrifuged at $200,000 \times g$ for 5 h at 4 °C (MLA80 rotor, Beckman-Coulter). The pellet of purified ribosomes was resuspended in storage buffer containing 50% glycerol, 50 mM Tris-HCl pH 7.5, 80 mM KCl, 10 mM $MgCl_2$, 0.5 mM EDTA, and 5 mM DTT.

## Mammalian cell extracts and ribosome purification

The whole cell extracts were prepared as follows: the cells were washed three times with ice-cold TBS buffer (50 mM Tris-HCl pH 7.4, 150 mM NaCl) and lysed with RIPA buffer (50 mM Tris-HCl pH 7.4, 150 mM NaCl, 1% NP-40, 0.1% SDS) directly on plates. Then, the cell lysates were transferred to 1.5 ml tubes, clarified by centrifugation at $12,000 \times g$ for 15 min at 4 °C, sonicated (two rounds of 10 sec-pulse with 30% amplitude), and centrifuged again with the same parameters as above. After clarification of the lysates, the protein concentration was measured using Bio-Rad Protein Assay Dye Reagent Concentrate (Bio-Rad, cat#5000006) and BSA (Merck, Sigma-Aldrich cat.#A4503) used for preparation of the standard curve.

Ribosomes from the MEF cell line were purified according to the procedure described in (Belin et al, 2010) with modifications. Briefly, the MEF cells were grown to about 75% confluency, washed three times with ice-cold TBS, harvested with a cell scraper, and spun down by centrifugation: $500 \times g$, 5 min at 4 °C. In the next step, the cells were resuspended in hypotonic buffer (50 mM

Tris-HCl pH 7.4, 250 mM sucrose, 250 mM KCl, 5 mM $MgCl_2$) containing 0.5 µM puromycin (Serva, cat#33835) and 1x complete Mini EDTA-free Protease Inhibitor Cocktail (Merck, Roche cat#5056489001) and incubated for 15 min on ice. Then, the cell homogenate was cleared in two steps. First, the nuclei were pelleted at $800 \times g$ for 10 min at 4 °C and, secondly, the mitochondria and other membrane fractions were sedimented using $16,000 \times g$ for 15 min at 4 °C. The obtained post-mitochondrial fraction was supplemented with KCl to the final concentration of 0.5 M, layered onto a sucrose cushion containing 0.5 M KCl, and centrifuged at $250,000 \times g$ for 3.5 h at 4 °C (MLA-80 rotor; Beckman-Coulter) to sediment the ribosomal fraction. The ribosomal pellet was washed with ice-cold MilliQ water and resuspended in storage buffer containing 50% glycerol, 50 mM Tris-HCl pH 7.5, 80 mM KCl, 10 mM $MgCl_2$, 0.5 mM EDTA, and 5 mM DTT.

## Polysome profiles of yeast cells

To obtain cell extracts for polysome profile analysis, the yeast cells were grown to $OD_{600} = 0.4–0.6$ with subsequent 20-min treatment with 100 µg/ml of cycloheximide (CHX, Merck, Sigma cat#C7698) and harvested by centrifugation. The cell pellet was resuspended in lysis buffer (10 mM Tris-HCl pH 7.5, 100 mM NaCl, 30 mM $MgCl_2$, 100 µg/ml CHX, 1 mM PMSF, 6 mM 2-mercaptoethanol, 1 nM pepstatin A, 10 nM leupeptin, 10 ng/ml aprotinin, 200 ng/ml heparin, and 1xRNase inhibitor (Merck, Sigma cat#R7397)) and the cells were disrupted by vigorous shaking with glass beads at 4 °C. The obtained cell lysate was cleared by centrifugation at $12,000 \times g$ for 10 min at 4 °C. The RNA content was measured at 260 nm and 15 U of RNA were run through a 10–50% sucrose gradient using a Beckman Coulter SW32 Ti rotor for 4.5 h at 26,500 RPM at 4 °C. The gradients were analyzed using an ISCO Brendel density gradient fractionator. The analyses of polysome profiles were conducted using MicrocalOrigin 6.0 software. The area of polysome fractions was calculated with the integration method to assess the total surface area and to determine the polysome-to-monosome (P/M) and 40S/60S ratios. For immunodetection of the fractions, proteins from each fraction were precipitated with TCA, centrifuged at $12,000 \times g$ for 30 min at 4 °C, washed twice with acetone, and analyzed by Western Blotting.

## Polysome profiles of mammalian cells and tissue

To prepare cell extracts for the polysome profile analysis, the cell lines were grown to about 75% confluency and the medium was refreshed 3 h before harvesting. Before harvesting, the cells were treated with cycloheximide (CHX, 100 µg/ml) (Merck, cat#C7698) for 3 min to stall the polysomal fraction. Next, the cells were washed twice with ice-cold DPBS (Gibco, cat#14040133) supplemented with 100 µg/ml of CHX and harvested by centrifugation at $500 \times g$ for 5 min at 4 °C. The pelleted cells were suspended in polysome lysis buffer (20 mM HEPES-KOH pH 7.4, 200 mM KCl, 5 mM $MgCl_2$, 0.5% NP-40, CHX 100 µg/ml) supplemented with EDTA-free Protease Inhibitor Cocktail (Roche, cat#5056489001) and RNase Inhibitor (Merck, cat#R7397) and incubated on ice for 30 min, with 15 s vortexing every 5 min. Then, the cell lysate was passed 20 times through a 23 G needle and incubated on ice for 5 min with 3 rounds of 1 min vortexing; the lysate was centrifuged at $10,000 \times g$ for 15 min at 4 °C. The RNA concentration was

measured at 260 nm, and six $OD_{260}$ units of cell extracts were loaded onto a 15–50% sucrose gradient (20 mM HEPES-KOH pH 7.4, 200 mM KCl, 5 mM $MgCl_2$) prepared on a gradient mixer (BioComp). The samples were centrifuged at 26,500 rpm using a Beckman Coulter SW32 Ti rotor for 4.5 h at 4 °C. The gradients were fractionated using an ISCO Brendel Density Gradient Fractionator, and the absorbance at 254 nm was recorded. Proteins in the collected fractions were precipitated overnight using TCA at 4 °C. The following day, the TCA-treated-fractions were centrifuged at $12,000 \times g$ for 20 min at 4 °C and then precipitated proteins were washed twice with ice-cold acetone and spun down at $12,000 \times g$ for 10 min at 4 °C. Dried protein pellets were resuspended in SDS-PAGE loading buffer, heated at 95 °C for 5 min, and analyzed by SDS-PAGE/Western Blotting.

The phosphatase-treated samples were prepared as follows: the cell lysate containing six $OD_{260}$ units of RNA was incubated with 70 U of FastAP Thermosensitive Alkaline Phosphatase (Thermo Fisher Scientific, cat#EF0652) for 1 h at 37 °C; lysate supplemented with FastAP buffer was used as a control. Both samples were processed following the protocol described above for the standard cell extracts used for the polysome profile analysis.

Male albino Swiss mice (weighing 20–30 g) obtained from a licensed breeder (Laboratory Animals Breeding, Ilkowice, Poland) were used for preparation of the polysomal fraction from mouse liver. The mice were adapted to the laboratory conditions for at least 1 week before the experiments. The animals were housed in groups of 8 per cage in controlled environmental conditions (temperature: 21–24 °C, relative humidity: 45–65%) with an artificial 12/12 h light/dark cycle (light on at 6:00 a.m.) and free access to food pellets and water. The housing and experimental procedures were conducted under the guidelines provided by the European Union Directive of 22 September 2010 (2010/63/EU) and Polish legislation concerning animal experimentation. All experimental procedures were approved by the Local Ethics Committee in Lublin, Poland (license no 107/2019). The freshly prepared liver was washed with 0.9% of ice-cold NaCl, cut into small pieces, weighted, and frozen in liquid nitrogen. Next, the liver was homogenized on ice in homogenization buffer (50 mM HEPES-KOH pH 7.4, 250 mM KCl, 5 mM $MgCl_2$, 250 mM sucrose, 100 µg/ml of CHX, 1 × RNase Inhibitor) using a glass-teflon Dounce homogenizer with 50 strokes. For 1 g of mouse liver, 3 ml of homogenization buffer was used. The tissue lysate was clarified by centrifugation at $3000 \times g$ for 15 min at 4 °C, supplemented with both 10% Triton X-100 and 13% sodium deoxycholate at a concentration of 100 µg/ml, and mixed gently. The RNA concentration was measured at 260 nm and ten $OD_{260}$ units were loaded onto a 15–50% sucrose gradient and separated using a Beckman Coulter SW32 Ti rotor at 26,500 rpm for 4.5 h at 4 °C. The gradient fractionation and analysis of polysomal fractions were performed as described above.

## Phos-tag analysis

Samples analyzed with the Phos-tag/Western Blotting method including purified ribosomes, total cell extracts, and cell extract used for polysome fractionation were prepared in buffers without EDTA. Phos-tag gels: 10% polyacrylamide supplemented with 50 µM Phos-tag (Wako Chemicals, cat#AAL-107) and 100 µM $ZnCl_2$ (Acros Organics) were run with the constant current of

25 mV for 3 h at 4 °C. Dephosphorylated and phosphorylated samples were used as controls. For dephosphorylation, 20 μg of ribosomes and 10 μg of total cell extracts were incubated for 1 h at 37 °C with 10 U of FastAP Thermosensitive Alkaline Phosphatase (ThermoFisher Scientific, cat#EF0652), whereas cell extracts (six $OD_{260}$ units of RNA) used for polysome fractionation were treated with 70 U of FastAP and processed analogously as the untreated samples until proteins from the polysome profile fractions were precipitated. Finally, all samples were diluted in SDS-PAGE loading buffer and heated for 5 min at 95 °C. Phosphorylated samples were prepared with CK2 kinase; to this end, 20 μg of purified ribosomes were incubated with 10 U of CK2 alpha subunit (kindly provided by Dr. K. Kubiński, KUL, Lublin, Poland) for 1 h at 37 °C, diluted in SDS-PAGE loading buffer, and boiled for 5 min at 95 °C. After electrophoresis, the gels were washed twice for 10 min with transfer buffer (25 mM Tris, 236 mM glycine, 20% methanol) supplemented with 10 mM EDTA pH 8.0, followed by washing in plain transfer buffer to remove EDTA. The transfer of proteins to PVDF membranes was carried out with a constant current of 300 mA for 3 h using wet-transfer, followed by immunodetection.

## Immunodetection

Protein samples (10 μg of cell extracts and 20 μg ribosomes) were resolved by SDS-PAGE electrophoresis in polyacrylamide gels (10% for the Phos-tag analysis and GCN2 detection and 12% for the other analyses) and transferred to PVDF membranes (Millipore, cat#IPVH00010) using a Mini Trans-Blot Cell Module (Bio-Rad). The membranes were blocked with 5% of non-fat dry milk in TBST buffer (50 mM Tris-HCl pH 7.4, 150 mM NaCl, 0.1% Tween-20) for 1 h with gentle orbital shaking at RT and, after washing with TBST, incubated with indicated primary antibodies diluted in TBST supplemented with 3% BSA for 2 h at RT. Then, the membranes were washed three times in TBST for 15 min followed by incubation with secondary antibodies diluted in TBST supplemented with 5% of non-fat milk for 1 h at RT. After three washes in TBST (15 min each), chemiluminescent signals were developed using ClarityTM Western ECL Substrate (Bio-Rad, cat#170-5061) and visualized with the aid of the Bio-Rad ChemiDocTM MP analyzer and Image Lab software (Bio-Rad). For quantitative analysis, chemiluminescent blot signals developed with horseradish peroxidase-conjugated secondary antibodies were acquired and analyzed with the system and software mentioned above and plotted using Microsoft Excel. For statistical analysis, one-sided Welch's two-sample t-tests were carried out at a fixed significance level of $\alpha = 0.05$. The tests were performed in the RStudio environment (R version 4.2.1) (http://www.rstudio.com).

## Antibodies

Antibodies directed toward tubulin (DM1A, #ab7291) and uL10 (#ab192866) were purchased from Abcam. Antibodies for eIF2α (D-3, #sc-133132) and uS7 (#sc-100832) detection were supplied by Santa Cruz Biotechnology. Antibodies for uL3 (#11005-1-AP) were purchased from Proteintech. Antibodies recognizing P1 (#HPA003368) were acquired from Atlas Antibody, and were used as a readout for the P1-P2 dimer. Antibodies for yeast uL10 and P1A/P2B were prepared in the Department of Molecular Biology, Lublin, Poland, based on purified recombinant proteins

(Tchorzewski et al, 1999) and were used as described previously (Boguszewska et al, 2002). Goat anti-rabbit IgG (H + L)-HRP Conjugate (#1706515) and goat anti-mouse IgG (H + L)-HRP Conjugate (#1706516) were purchased from Bio-Rad. Antibodies recognizing Gcn2 kinase were kindly provided by Alan Hinnebusch (National Institutes of Health, Bethesda, MD, USA). Antibodies against uL23 were kindly provided by Ed Hurt (University of Heidelberg, Germany).

## Meta-analysis of proteomic data for P-stalk protein phosphorylation

The PeptideAtlas platform was used to map the ribosomal P-stalk protein phosphorylation (Desiere et al, 2006). The Human 2022-01-build was used to search for experimentally observed peptides with the C-terminal group and two C-terminal serine residues. The following dataset was used: build name—human 2024-01, reference_database—THISP_2024-01-01 + gEVE1.1 + TEs, with distinct_peptides 3,945,444, #Experiments 3160, canonical_proteins 17,416. Once the list was derived, the peptides with the highest number of observations (spectra) were selected for each protein. The VEAKEESEESDEDMGFGLFD peptide was used for the human protein uL10, whereas the single common peptide with the sequence KEESEESDDDMGFGLFD was used for the human P1 and P2 proteins. The phosphorylation observed within the peptide was classified into four groups: first serine residue modified calculated from NTD (S304, S101, and S102, for uL10, P1, and P2, respectively), second serine residue modified (S307, S104, and S105 for uL10, P1, and P2, respectively), both serine residues modified, or no serine residue modified. The peptides were searched in the Human Phosphoproteomics 2022-04 build. In the 2022-04 build, the phosphorylation observed in the peptides was divided into three groups; first, second, and both serine residues modified.

## Isolation and in vitro dephosphorylation of the ribosomal P-stalk

The purification of the P-stalk complex was performed using *S. cerevisiae* uL10 mutant uL10TH199 and the procedure described previously (Grela et al, 2010). In brief, the mutant strain was used where uL10 has a short amino acid sequence recognized by thrombin placed at position 199 with 6xTag, and upon thrombin treatment the truncated uL10 protein (amino acids 199–312) can be released with P1A-P2B and P1B-P2A dimers ($\Delta uL10_{199-312}$-(P1A–P2B)-(P1B–P2A)). Thus, to release the pentameric P-stalk complex, 80S ribosomal particles were isolated from P0TH199 yeast cells and treated with activated thrombin from bovine plasma (Sigma, # T4648-1KU) (1 U of thrombin/10 mg of 80S) for 12 h at 4 °C. In vitro P-stalk dephosphorylation was performed during the ribosome isolation procedure. Briefly, the ribosomes purified with the sucrose cushion were treated with AP according to the protocol applied for the preparation of Phos-tag samples described above. The wild-type and AP-dephosphorylated ribosomes were ultracentrifuged ($100,000 \times g$) and resuspended in buffer (10 mM Tris pH 7.4, 150 mM NaCl, 10 mM $MgCl_2$), and the thrombin-cleaved P-stalk complexes were purified to homogeneity using affinity (Ni-NTA agarose from QIAGEN) and size-exclusion chromatography (Superose 12, 12/300HR column; GE Healthcare). The purity of the

ribosomal stalk complexes was verified by 12% SDS-PAGE, 10% native PAGE, and native mass spectrometry.

## Native mass spectrometry of P-stalk complexes

All complexes were analyzed using a SYNAPT G2-Si High-Definition Mass Spectrometer (Waters, Manchester, UK). For the analysis, all protein solutions were buffer-exchanged into 200 mM ammonium acetate (pH 7.5) using Micro Bio-Spin chromatography columns (Bio-Rad). Aliquots of 2 µl were introduced into the mass spectrometer via nanoflow capillaries, and the following conditions were applied: capillary voltage 1.2 kV, sampling cone 120 V, and source offset 20 V. The source temperature was set at 25 °C. The collision voltage was adjusted for an optimal signal level. Maximum entropy (MaxEnt, Waters) deconvolution was applied to the electrospray data to recalculate the gas phase-existing masses according to previously published results (Grela et al, 2010).

## Microscale thermophoresis (MST)

Purified P-stalk complexes were fluorescently labeled according to the manufacturer's instructions with a Monolith His-Tag Labeling Kit RED-tris-NTA 2nd Generation kit (NanoTemper Technologies, GmbH, Germany). The concentration of the labeled P-stalk protein complexes was adjusted to 5 nM. A dilution series of unlabeled eIF5B protein was prepared in a final volume of 20 µl for the MST measurements, starting from the highest 1.48 µM concentration in PBS buffer with 0.05% Tween-20 and 1 mg/ml of bovine serum albumin. Thermophoresis was performed using a Monolith NT.115 Pico instrument (NanoTemper Technologies) at an ambient temperature of 25 °C and 1/14/1 s laser off/on/off cycles, respectively. The instrument parameters were adjusted to 100% excitation power and a medium MST detection range. The data were analyzed using NanoTemper NT analysis software (version 2.2.4) at the standard MST time of 30 s (thermophoresis plus T-jump). The $K_d$ of the interaction was determined by fitting the data using fit algorithms implemented into the MST analytical software. To calculate the fraction bound, the ΔFnorm value of each point was divided by the amplitude of the fitted curve, resulting in values from 0 to 1 (0 = unbound, 1 = bound).

## Assays of dipeptides and tripeptides and GTP hydrolysis

The rate of the first and second peptide bond formation was determined from time courses using the 80S initiation complex (ICs) prepared with the wild-type WT (phosphorylated) or dephospho-mimicking ribosomes (SDPM) as described previously (Ranjan et al, 2021). The integrity of ribosomal subunits was verified by mass-spectrometry analysis of ribosomal proteins (r-proteins), subjected to trypsin digestion (Appendix Table S1). To form the ternary complex, 8 µM of eIF2 was incubated with 1 mM GTP and 4 µM [³H]Met-tRNA$_i^{Met}$ in YT buffer (30 mM HEPES-KOH pH 7.5, 100 mM KOAc, 3 mM MgCl₂) at 26 °C for 15 min. To form 80S IC, 2 µM 40S, 10 µM mRNA, 10 µM eIF-mix (mixture of initiation factors eIF1, eIF1A, eIF3, eIF5), 2 mM DTT, 0.25 mM spermidine, and 1 mM GTP were incubated for 5 min at 26 °C

before adding 3 µM 60S subunits and 6 µM eIF5B, followed by addition of the ternary complex with simultaneous adjustment of the MgCl₂ concentration to 9 mM (YT9 buffer). 80S ICs were purified by ultracentrifugation through a 1.1 M sucrose cushion, and the pellets were dissolved in YT9 buffer. 80S ICs were diluted to 3 mM MgCl₂ prior to the experiments. Ternary complexes eEF1A–GTP–[¹⁴C]Phe-tRNA$^{Phe}$ and eEF1A–GTP–[¹⁴C]Val-tRNA$^{Val}$ were prepared by incubating 1 µM eEF1A, 1 µM eEF1Bα, 3 mM PEP, 1% PK, 1 mM DTT, and 2 mM GTP in YT buffer for 15 min at 26 °C. 0.2 µM [¹⁴C]Phe-tRNA$^{Phe}$ or [¹⁴C]Val-tRNA$^{Val}$ (5 eEF1A:1 aa-tRNA) was then added and incubated for additional 5 min at 26 °C, followed by addition of 2 µM modified eIF5A. For each time point, 0.7 µM of the pre-formed 80S IC with [³H]Met-tRNA$_i^{Met}$ in the P site was rapidly mixed with 0.2 µM eEF1A–GTP–[¹⁴C]Phe-tRNA$^{Phe}$ in the quench-flow apparatus. After the desired incubation time, the reaction was quenched with 0.5 M KOH and the dipeptides were analyzed by Reverse-Phase High Performance Liquid Chromatography (HPLC) following [³H] and [¹⁴C] counts (Wohlgemuth et al, 2008). For the analysis of tripeptide formation, 0.7 µM of the initiation complexes ([³H]Met-tRNA$^{Met}$) was mixed with 1 µM eEF1A–GTP–Phe-tRNA$^{Phe}$ and incubated for 15 min at 26 °C to form dipeptides. The dipeptides were then mixed with 0.7 µM eEF1A–GTP–[¹⁴C]Val-tRNA$^{Val}$, 2 µM eEF2, 4 µM eEF3 and with 40 µM ATP, in the quench-flow apparatus and the reaction was quenched with KOH after the desired time point. The samples were further processed as for the dipeptide analysis. The mRNA used for all analyses was purchased from IBA Lifesciences GmbH, with sequence as follows: 5′-GGUCUCUCUCUCUCUCUCUAUGUUU GUUUCUCUCUCUCUC-3′.

TrGTPases eIF5B, eEF1A, and eEF2 as well as ribosomes were purified as previously described (Blanchet and Ranjan, 2022). GTP hydrolysis reactions were performed by mixing 0.05 µM GTPases with 10 µM [γ³²P]GTP with 0.2 µM wild-type ribosomes or ribosomes from the SDPM mutant strain mimicking the non-phosphorylated states of the ribosomes. The reactions were quenched with 50% formic acid (25% final). The amount of hydrolyzed [γ³²P] GTP was analyzed by thin-layer chromatography using Polygram CEL300 TLC plates (Gromadski and Rodnina, 2004). The [Pi/(Pi + GTP)] ratio was used to estimate the time dependence of the GTP hydrolysis.

## Quantification of misincorporation errors using the dual-luciferase assay

The efficiency of misincorporation was measured using the dual-luciferase assay according to an established procedure (Harger and Dinman, 2003) with modifications described earlier (Wawiorka et al, 2017). Briefly, each yeast strain was transformed with a control plasmid or with appropriate vectors: for misincorporation, the pDB688 vector was used as a control; pDB868, with a near-cognate CGC$_{245}$ codon, having replaced a cognate CAC codon encoding the His amino acid at position 245 of the firefly luciferase with a CGC codon for Arg; all the genetic constructs were kindly provided by Dr David Bedwell (Salas-Marco and Bedwell, 2005). The pJD375 vector was also used as a control plasmid with cognate AGA at position 218 (coding Arg) of the firefly luciferase, and pJD643 with near-cognate AGC$_{218}$ coding Ser, provided as a kind gift by Dr Jonathan Dinman (Plant et al, 2007).

## Determination of the yeast lifespan

The total yeast cell lifespan is calculated as the sum of the reproductive (i.e., time between the first and last budding) and post-reproductive (time after the last budding until death) lifespan. The total lifespan of the *S. cerevisiae* yeast was determined as previously described (Minois et al, 2005) with further modification (Molon and Zebrowski, 2017). A fresh logarithmic yeast culture ($OD_{600} = 0.6–0.8$) was collected and transferred to YPD plates with solid medium containing Phloxine B (10 µg/ml). During manipulation, the plates were kept at 28 °C for 15 h and at 4 °C during the night. The results represent measurements for at least 100 cells analyzed in at least three independent experiments. The analysis was performed by micromanipulation using the Nikon Eclipse E200 optical microscope with an attached micromanipulator. The budding lifespan of the yeast cells was assessed as follows: yeast cells were cultured on YPD medium plates after an overnight growth period using a micromanipulator. The budding lifespan was determined by microscopic examination, following a known procedure (Molon and Zebrowski, 2017). The number of buds produced by each mother cell served as an indicator of the budding lifespan. A minimum of 90 cells were meticulously examined in two independent experiments. The analysis was performed using a Nikon Eclipse E200 optical microscope (Nikon, Amsterdam, Netherlands) equipped with an attached micromanipulator. Statistical significances were assessed using ANOVA and Dunnett's post hoc test (*$p < 0.05$). Translational efficiency was measured as incorporation of $^{35}$S-methionine, and the half-transit time was calculated according to previously described protocols (Wawiorka et al, 2017).

## Ribo-seq

For ribosome profiling, yeast cells (BY4741 and SDPM strains) were grown in synthetic complete medium supplemented with glucose (2%) and inositol (200 mM). 1 l of the medium was inoculated to $OD_{600}$ of 0.06–0.08 from a saturated starting culture. The cells were grown to $OD_{600}$ of 0.4–0.5 and treated with tunicamycin at a concentration of 2.5 µg/ml. The cells were harvested by fast filtration and frozen in liquid nitrogen. Isolation of ribosome-protected footprints and generation of libraries for Illumina sequencing are described in detail elsewhere (Shafieinouri et al, 2022). Analysis of RNA-seq and Ribo-seq Data was performed as follows: FastQC reports were employed to assess the quality of the raw data. Following this initial step, TrimGalore software was used to remove adapters and low-quality sequences. The Bowtie tool was applied to eliminate sequences corresponding to tRNA and rRNA based on genome reference data (RefSeq: GCF000146045.2). After the preprocessing steps, the FastQC tool was reapplied for quality control (www.bioinformatics.babraham.ac.uk/projects/fastqc/). Read length distribution plots were generated to analyze the distribution of read lengths in the Ribo-seq data. The trimmed data were then mapped to the genome using STAR, allowing for a tolerance of up to 4% mismatch in the read length. Feature Counts was employed to count uniquely mapped reads, utilizing gene annotations from the RefSeq database. Subsequently, read count normalization was carried out using the transcripts per million (TPM) method specifically for coding sequences (CDS). For the Ribo-seq data, codon analysis based on 21 nt and 28 nt ribosome protected fragments (RPFs) and ribosome occupancy on CDS were assessed using RiboToolkit (Liu et al, 2020).

## Molecular dynamics simulations

For MD simulation, initial structures were built using the UCSF ChimeraX tool (Goddard et al, 2018) as a linear chain or an α-helix. The MD simulations were performed using the GROMACS (Abraham et al, 2015) package with the CHARMM36 (Best et al, 2012) all-atom force field and the TIP3P (Jorgensen et al, 1983) water model. Each peptide was solvated in a periodic box with a 0.9 nm distance between the peptide and the box walls and neutralized using sodium ions. All the simulations were performed in the following steps: energy minimization was performed using the steepest descent algorithm until forces have converged to within the available machine precision; system heating was carried out for 500 ps, from 0 to 298 K, in NVT conditions (canonical ensemble); system equilibration was performed for 1000 ps, at 300 K and 1.0 bar, in NPT conditions; production was performed for at least 2 µs, with the time step set to 2 fs at constant pressure 1 bar and constant temperature 298 K. All the simulation analyses presented in this paper were performed using GROMACS package tools, the DSSP program, and the Python programming language. Structural visualization and graphical representation of data were rendered using the UCSF ChimeraX tool and Origin program [Origin, Version 2022b. OriginLab Corporation, Northampton, MA, USA]. Clustering was carried out using the Gromos clustering algorithm (Daura et al, 1999) with a 0.3 nm clustering cut-off. Three-dimensional Free Energy Landscapes (FELs) were generated by trajectory projection on eigenvector 1 and eigenvector 2 (PC1 and PC2) (Campos and Baptista, 2009).

# Data availability

The RNA-seq and Ribo-seq data have been deposited in NCBI's SRA database (Sayers et al, 2022) and are accessible through BioProject accession number: PRJNA1071838 (https://www.ncbi.nlm.nih.gov/bioproject/PRJNA1071838).

The source data of this paper are collected in the following database record: biostudies:S-SCDT-10_1038-S44319-024-00297-1.

# Peer review information

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

## Acknowledgements

We are grateful to Emilia Łabuć for excellent technical assistance in data processing. We thank Tessa Hübner, Olaf Geintzer, and Thomas Shulz for expert technical assistance. This work was supported by grants from the National Science Center in Poland [UMO-2018/29/B/NZ1/01728] to MT; 2018/30/E/NZ1/00605 to DK; 2022/45/B/NZ3/02353 to PG, as well as by the Leibniz Prize and SFB 1565 (project number 469281184) of the Deutsche Forschungsgemeinschaft and the Max Planck Society to MVR.

## Author contributions

**Kamil Filipek**: Investigation; Visualization; Methodology; Writing—original draft; Writing—review and editing. **Sandra Blanchet**: Investigation; Visualization; Methodology; Writing—original draft; Writing—review and editing. **Eliza Molestak**: Investigation; Methodology; Writing—review and editing. **Monika Zaciura**: Investigation; Methodology; Writing—review and editing. **Colin Chih-Chien Wu**: Data curation; Investigation. **Patrycja Horbowicz-Drożdżal**: Investigation; Visualization; Writing—review and editing. **Przemysław Grela**: Data curation; Funding acquisition; Investigation; Visualization; Methodology; Writing—original draft; Writing—review and editing. **Mateusz Zalewski**: Software; Investigation; Visualization; Methodology; Writing—original draft; Writing—review and editing. **Sebastian Kmiecik**: Data curation; Investigation; Methodology; Writing—original draft; Writing—review and editing. **Alan González-Ibarra**: Investigation. **Dawid Krokowski**: Data curation; Funding acquisition; Writing—review and editing. **Przemysław Latoch**: Methodology; Writing—review and editing. **Agata Starosta**: Methodology; Writing—review and editing. **Mateusz Mołoń**: Investigation; Writing—review and editing. **Yutian Shao**: Investigation. **Lidia Borkiewicz**: Investigation. **Barbara Michalec-Wawiórka**: Investigation; Visualization; Writing—review and editing. **Leszek Wawiórka**: Conceptualization; Data curation; Writing—original draft; Writing—review and editing. **Konrad Kubiński**: Investigation; Writing—review and editing. **Katarzyna Socala**: Resources. **Piotr Wlaz**: Resources. **Kyle W Cunningham**: Conceptualization; Writing—original draft; Writing—review and editing. **Rachel Green**: Conceptualization; Data curation; Writing—original draft; Writing—review and editing. **Marina V Rodnina**: Conceptualization; Data curation; Funding acquisition; Writing—original draft; Writing—review and editing. **Marek Tchorzewski**: Conceptualization; Data curation; Supervision; Funding acquisition; Visualization; Writing—original draft; Project administration; Writing—review and editing.

Source data underlying figure panels in this paper may have individual authorship assigned. Where available, figure panel/source data authorship is listed in the following database record: biostudies:S-SCDT-10_1038-S44319-024-00297-1.

## Disclosure and competing interests statement

The authors declare no competing interests.

# Expanded View Figures

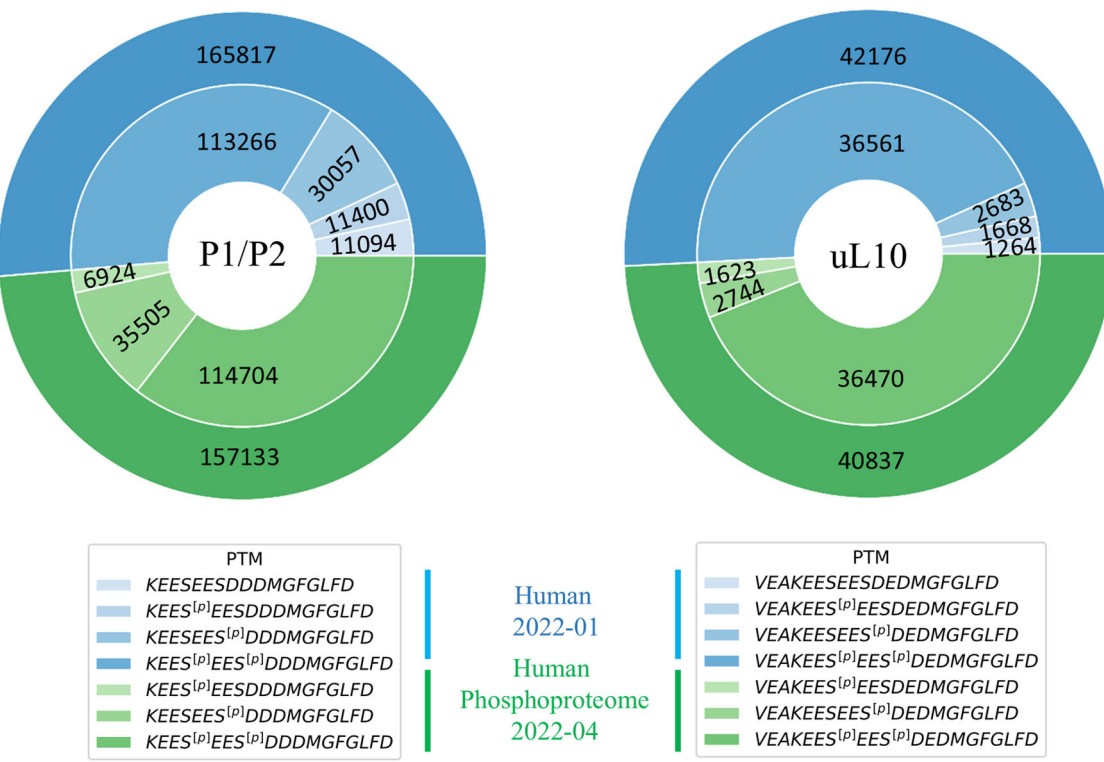

**Figure EV1.   Meta-analysis of CTD P-stalk protein phosphorylation in mammalian cells using mass spectrometry data implemented in the PeptideAtlas database.**

The nested pie charts show the identified peptides of the P1/P2 (left chart) and uL10 (right chart) proteins. The upper, blue part of the plots represents the observation from the entire Human Proteome 2022-01 assembly. The lower, green part of the plots shows the results from the Human Phosphoproteome 2022-04 build. The Human Phosphoproteome 2022-04 contains only phosphorylated P1/P2 and uL10 proteins. The complete Human Proteome 2022-01 includes the observation for non-phosphorylated P-stalk proteins. The total number of peptide observations is shown in the outer ring of the plots. The inner ring shows the number of results with different patterns of serine residue phosphorylation. The higher the number of observations, the darker the color (blue or green). The lower panel shows the identified peptides within the two datasets, 2022-01 and 2022-04 builds, in the corresponding colors; phosphorylation site is marked with [p].

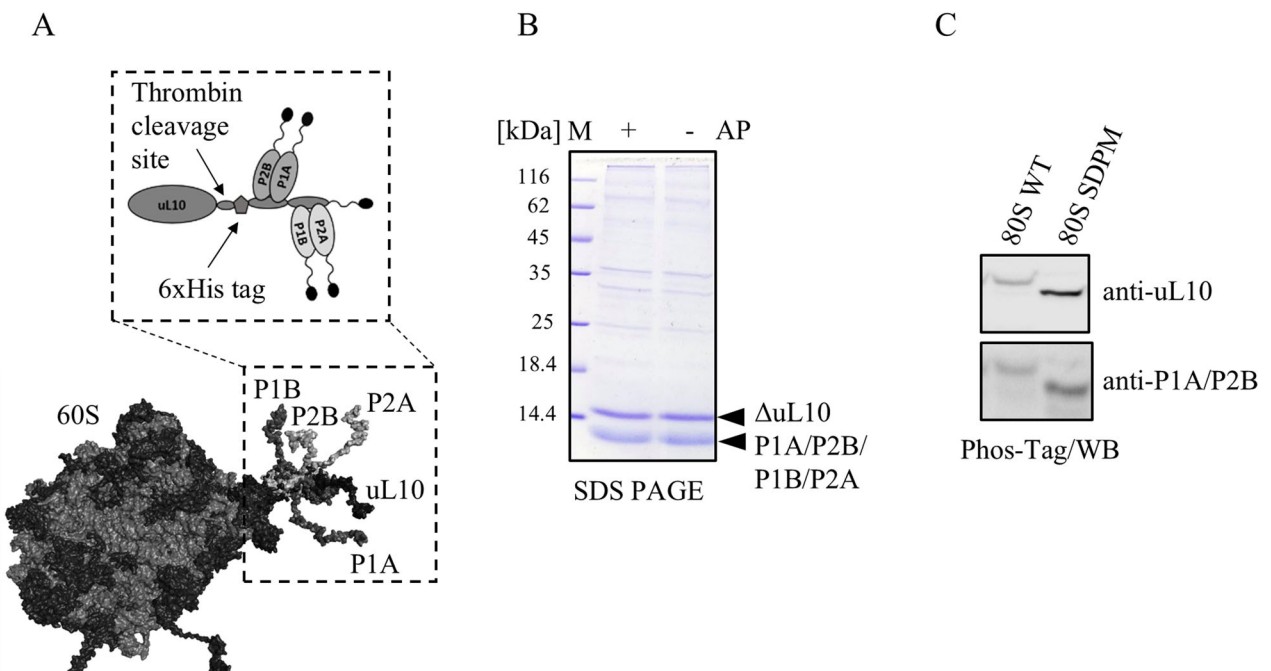

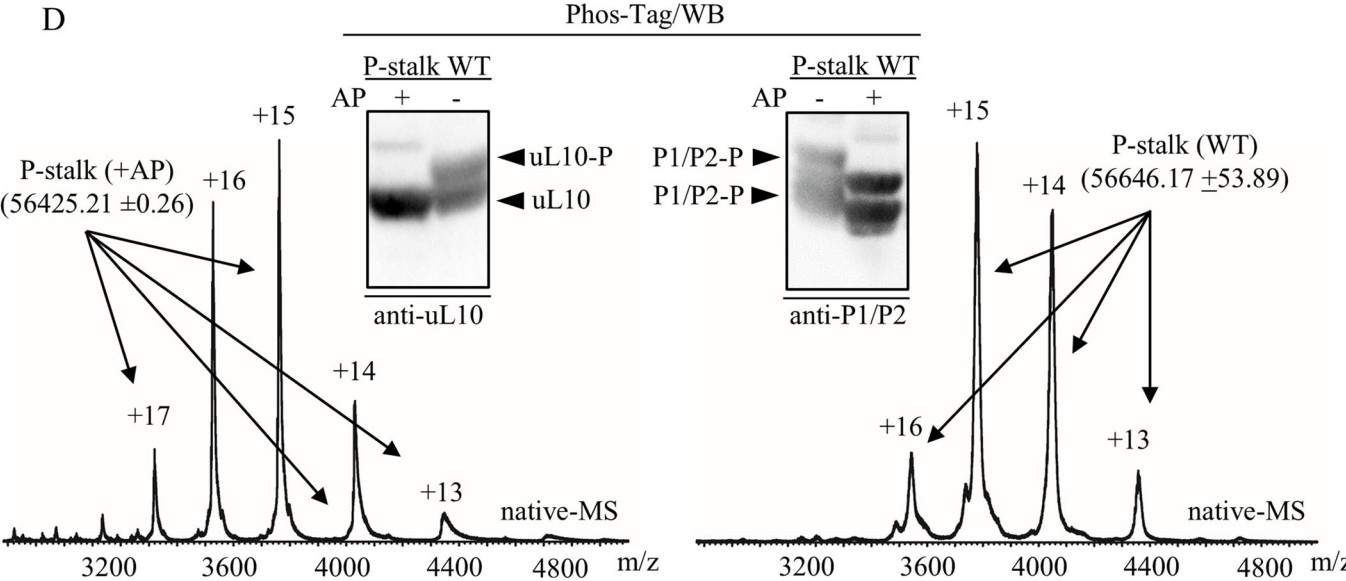

**Figure EV2. Purification of phosphorylated (native) and unphosphorylated (AP-treated) P-stalk complexes and 80S ribosomal particles from WT and SDPM mutant strains.**

(A) Schematic representation of the genetically engineered ribosomal particles with the P-stalk scheme. Specific thrombin cleavage site and 6xHis-tag used for P-stalk release and subsequent purification are indicated with arrows. (B) SDS-PAGE analysis of purified native (-AP) and dephosphorylated (+AP) stalk complexes. The positions of ΔuL10 and P1A/P2B/P1B/P2A are pointed with arrows. (C) Phos-tag analysis of purified 80S ribosomal particles; 80S WT - ribosomes from WT strain, 80S SDPM - ribsomes from SDPM mutant strain harboring S to A mutations within all P-protein; anti-uL10 and anti-P1A/P2B antibodies were used. (D) Native mass spectroscopy analysis (native-MS) of purified intact P-stalk complexes; left and right panels, MS spectra of the complexes of WT and +AP, respectively; numbers next to the peaks indicate the charge states of the complexes; molecular masses of the P-stalk complexes were calculated by MaxEnt deconvolution software (Waters), and are 56646 and 56425 Da for WT and +AP complexes, respectively; insert - phos-tag analysis of purified stalk complexes, WT and +AP. The complexes were analyzed by SDS-PAGE/ phos-tag/WB and subsequently proteins were detected with specific antibodies against uL10 and P1/P2 proteins. The positions of phospho- and dephospho-forms are indicated with arrows.

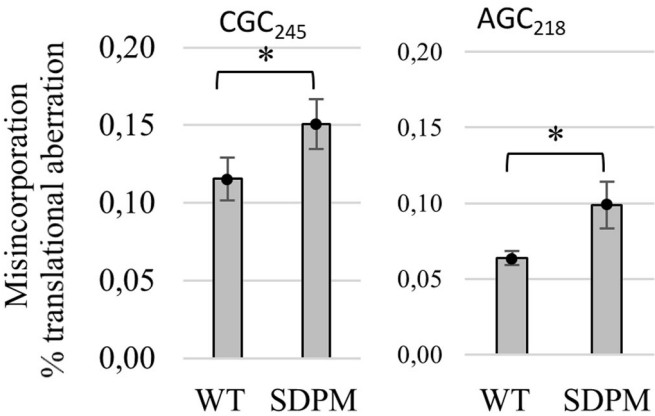

**Figure EV3.   Misincorporation analysis using a dual-luciferase reporter assay.**

CGC$_{245}$ and AGC$_{218}$ describe near-cognate codons at positions 245 and 218 of the firefly reporter enzyme. All data are presented as the percentage of translational aberration; error bars, standard deviations ($n = 3$, technical replicates); *$p < 0.05$ by Student's *t*-test is indicated by asterisks.

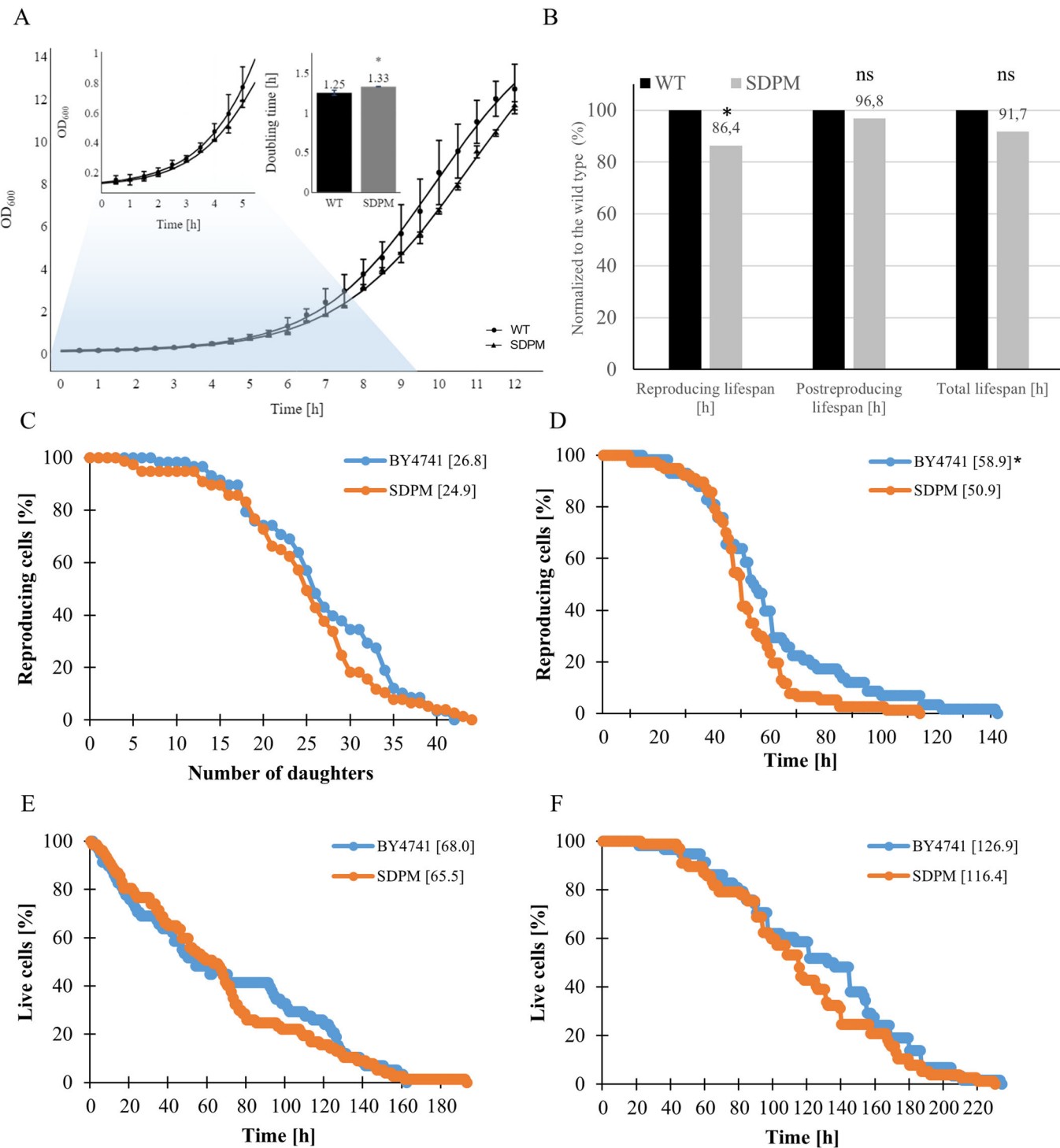

**Figure EV4. Cellular fitness of the WT and SDPM yeast strains.**

(A) Growth curves for WT and SDPM strains. The exponential fragment of growth curves used for doubling time calculation and average doubling time for WT and SDPM are shown in insets. The statistical analysis was done using one-tailed t-Welch test (*$p < 0.05$), data are presented as mean ± SEM of $n = 3$, technical replicates. (B) Yeast lifespan analysis of WT and SDPM strains on the single cell level; the values were normalized to the WT referred as 100%, using data presented in (D–F). To assess differences between the WT and SDPM strains, one-way ANOVA and Dunnett's post hoc tests were used (*$p < 0.05$, ns not significant. Comparison of the reproductive potential (A), reproductive lifespan (B), post-reproductive lifespan (C), and total lifespan (D) of the haploid reference yeast strain BY4741 (wild-type—WT) and the mutant SDPM strain. Statistical significances were assessed using ANOVA and Dunnett's post hoc test (*$p < 0.05$). The mean value for a total of 90 cells from two independent experiments is shown in parentheses.

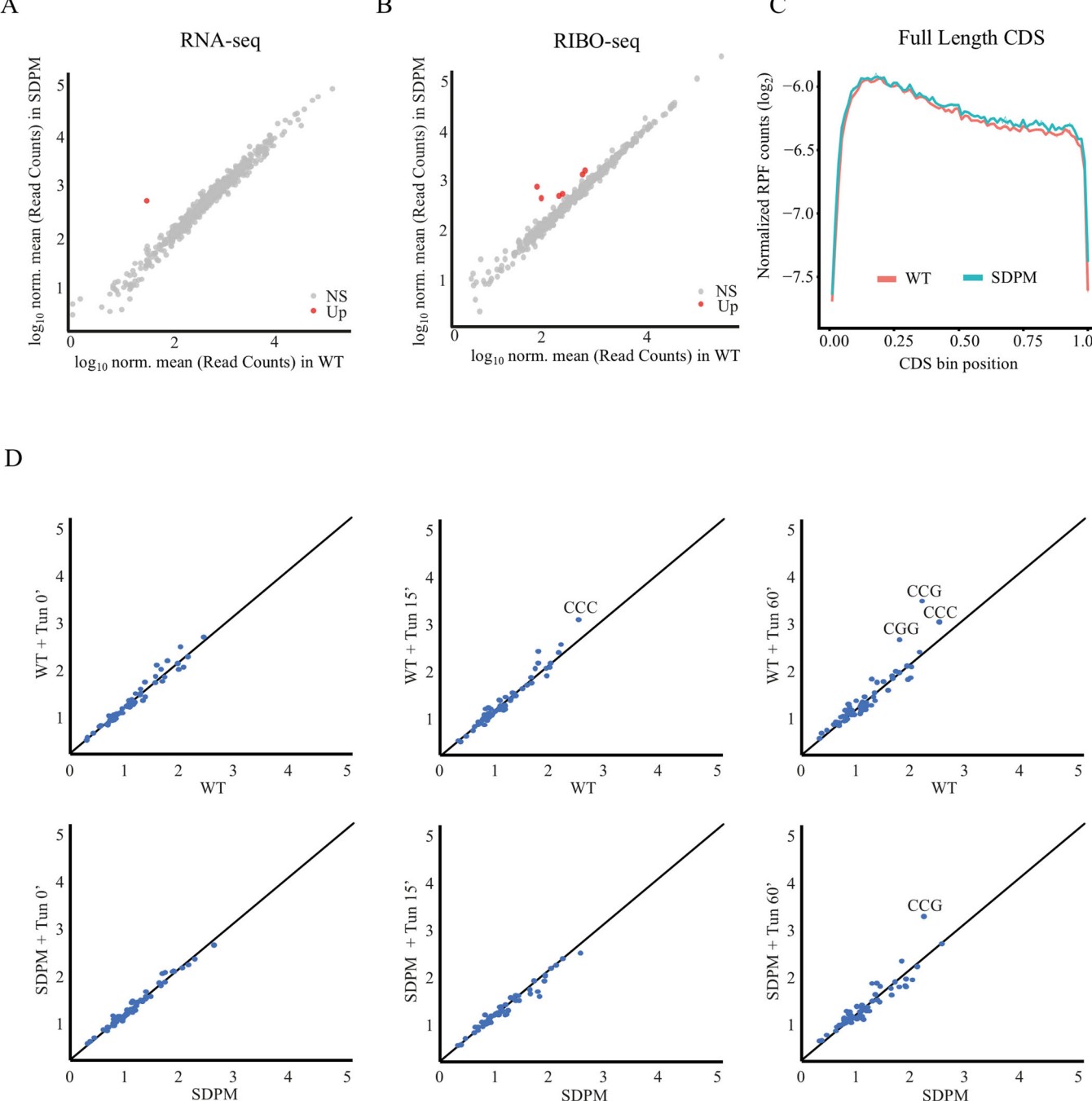

**Figure EV5. The ribosomal footprints distribution obtained based on a genome-wide scale analysis of WT and SDPM yeast strains.**

(A, B) Cross-correlation of gene expression analyses at the level of transcription (RNA-seq - A) and translation (RIBO-seq - B) for WT vs SDPM mutant strains; (C) average ribosome occupancy from all genes aligned from start to stop codons within the coding sequence (CDS) for WT (red line) and SDPM mutant strain (blue line); ribosome occupancy was normalized to show a mean value of 1 for each codon; the footprint occupancy was shown for the whole CDS. (D) The correlation of specific codon occupancies within 28 nt RPF. Specific overrepresented codons, CGG for arginine and CCG, CCC for proline are indicated.

