## [Peer Review File · EMBO Reports]

Phosphorylation of P-stalk proteins defines the ribosomal state for interaction with auxiliary protein factors

Kamil Filipek, Sandra Blanchet, Eliza Molestak, Monika Zaciura, Colin Chih-Chien Wu, Patrycja Horbowicz-Drożdżał, Przemysław Grela, Mateusz Zalewski, Sebastian Kmiecik, Alan González-Ibarra, Dawid Krokowski, Przemysław Latoch, Agata Starosta, Mateusz Mołoń, Yutian Shao, Lidia Borkiewicz, Barbara Michalec-Wawiórka, Leszek Wawiórka, Konrad Kubiński, Katarzyna Socala, Piotr Wlaz, Kyle W. Cunningham, Rachel Green, Marina Rodnina, and Marek Tchorzewski

Corresponding author(s): Marek Tchorzewski (marek.tchorzewski@mail.umcs.pl), Kyle W. Cunningham (kwc@jhu.edu), Rachel Green (ragreen@jhmi.edu), Marina Rodnina (rodnina@mpinat.mpg.de)

Review Timeline:

Submission Date:	22nd Apr 24
Editorial Decision:	16th May 24
Revision Received:	31st Jul 24
Editorial Decision:	2nd Sep 24
Revision Received:	3rd Oct 24
Accepted:	14th Oct 24

Editor: Esther Schnapp

Transaction Report:

Dear Prof. Tchorzewski,

Thank you for the submission of your manuscript to EMBO reports. We have now received the full set of referee reports that is pasted below.

As you will see, the referees acknowledge that the findings are potentially interesting. However, they also have some concerns and several suggestions for how the study could be improved. I think all points are reasonable and should be addressed. Please let me know in case you disagree, and we can discuss the exact revision requirements further, also in a video chat, if you like. Please note though that I will not be in the office the following 2 weeks, but I will be back on the 3rd of June.

I would thus like to invite you to revise your manuscript with the understanding that the referee concerns must be fully addressed and their suggestions taken on board. Please address all referee concerns in a complete point-by-point response. Acceptance of the manuscript will depend on a positive outcome of a second round of review. It is EMBO reports policy to allow a single round of major revision only and acceptance or rejection of the manuscript will therefore depend on the completeness of your responses included in the next, final version of the manuscript.

We realize that it is difficult to revise to a specific deadline. In the interest of protecting the conceptual advance provided by the work, we recommend a revision within 3 months (16th Aug 2024). Please discuss the revision progress ahead of this time with the editor if you require more time to complete the revisions.

- 1) A data availability section providing access to data deposited in public databases is missing. If you have not deposited any data, please add a sentence to the data availability section that explains that.
- 2) Your manuscript contains statistics and error bars based on $n=2$. Please use scatter blots in these cases. No statistics should be calculated if $n=2$.

3) We replaced Supplementary Information with Expanded View (EV) Figures and Tables that are collapsible/expandable online. A maximum of 5 EV Figures can be typeset. EV Figures should be cited as 'Figure EV1, Figure EV2' etc... in the text and their respective legends should be included in the main text after the legends of regular figures.

5) a complete author checklist, which you can download from our author guidelines <https://www.embopress.org/page/journal/14693178/authorguide>. Please insert information in the checklist that is also reflected in the manuscript. The completed author checklist will also be part of the RPF.

6) Please note that all corresponding authors are required to supply an ORCID ID for their name upon submission of a revised manuscript (<https://orcid.org/>). Please find instructions on how to link your ORCID ID to your account in our manuscript tracking system in our Author guidelines <https://www.embopress.org/page/journal/14693178/authorguide#authorshipguidelines>

- the name of the statistical test used to generate error bars and P values,
- the number (n) of independent experiments (please specify technical or biological replicates) underlying each data point,
- the nature of the bars and error bars (s.d., s.e.m.),
- If the data are obtained from $n < 2$, use scatter blots showing the individual data points.

I look forward to seeing a revised form of your manuscript when it is ready.

Referee #1:

EMBOR-2024-59463-T

In this manuscript Filipek et al use a combination of growth assays, biochemical analyses, translation assays and molecular dynamics simulations to decipher a role for phosphorylation of the stalk proteins. The work is carefully executed and of interest. The data support the model. I have a few suggestions to improve the impact of the work, and increase confidence in the conclusions.

1. The differences in binding of eIF5B to the phosphorylated and dephosphorylated stalk is likely an underestimate, because some of the phosphorylated stalk is dephosphorylated. The authors should note this.
2. It would be great to see at least an SDS-Page gel (or better 2D gel, or mass-spec data) to confirm that the wt and SDMP ribosomes are otherwise identical.
3. The authors carry out a very systematic analysis of effects arising from the phosphor-deficient SDMP mutant. I'd be curious to know if the phosphomimetic mutant has any phenotypes, especially under stress, or conditions that lead to open A-sites.
4. I would suggest the authors view everything in relation to the wt strain. E.g. p. 18, l. 10: the codon occupancy was decreased in the mutant...
5. The suggestions from the MD should be tested by mutagenesis of the lysine residues. Their effects can be tested in the in vivo assays that give a phenotype for the SDPM mutant.

Referee #2:

In the submission by Filipek et al., the authors test the hypothesis that differential phosphorylation levels of the eukaryal P-stalk protein uL10 regulates and possibly interconnects both the ribosome's translational activity (by affecting recruitment/activation of translational GTPases) with the Integrated Stress Response (by activation of the stress-related Gcn2 kinase). Given their major findings that P-stalk proteins are exclusively in the phosphorylated state and that the phosphorylation level of the P-stalk is constant and the fact that protein biosynthesis activities of the translational machinery are largely unaffected, the hypothesis does not seem to be supported by the obtained data.

The presented data are (most of the time) of high quality and are thus relevant for falsifying the starting hypothesis. There are several points that merit consideration:

- a) Introduction, page 3: The authors should mention that the ribosomal P-stalk architecture also involves uL11 (beside uL10). It has been shown in yeast and human system that uL11 contributes to translation dynamics.
- b) Fig. 4A: The authors state in the Results that "The AP treatment of the isolated cell extracts prior to fractionation did not affect the overall shape of polysomal profiles in the yeast (Fig. 4A, left) or mammalian (Fig. 4A, right) cells." This statement is not supported by the polysome profile in mammalian cells since especially the heavy polysomes were significantly altered upon AP treatment.
- c) Fig. 4B: The quality of the Phos-tag western blot for the yeast sample is quite poor which affects the conclusions that the authors draw from these experimental data. e.g. the bands in the last fraction #9 are half cut.
- d) Fig. S3: it has been shown before that the dual luciferase reporter used here, which relies on the misincorporation of amino acids at the His245 codon, is not ideal in the yeast system due to its very high background activity for many amino acids incorporated at that position within the active site. It has been demonstrated that the Lys529 mutant is superior due to its low background signal.
- e) Given their main findings (phosphorylation status of the P-stalk is unaltered and has little or no effect on growth as well as on the overall performance of the translational machinery; e.g. first sentence on page 16; or abstract, etc), the wording "Phosphorylation....modulates ribosome functions" or even the title of the manuscript "Phosphorylation fine-tunes ribosomal-factor interactions" is misleading. A reader could anticipate that words such as "fine-tuning" or "modulation" would imply that altered in vivo levels of P-stalk protein phosphorylation would adjust ribosomal activities. The authors however demonstrate in this study that both are actually not happening. Thus I strongly suggest re-considering such statements.
- f) The effects of tunicamycin are typically seen after several hours (once the ER gets too crowded and the UPR gets activated). The observed response after solely 15 min but not after 60 min is therefore surprising (Fig. 8A). Are cells already stressed after 15 min tunicamycin treatment? Could the authors use e.g. eIF2alpha phosphorylation or activation of any of the known ER-stress sensors to convince the reader of an indeed activated stress response? The data shown in Fig. 9A do not appear to suggest stress activation after 15 min Tun treatment. This should be clarified.
- g) The authors state that Gcn2 accumulates on the 60S ribosomal subunit in the WT strain upon Tun treatment, but not in the SDPM cells. Based on the blot shown in Fig. 9B such a statement does not seem to be justified given the fact that the western blot signal for Gcn2 in the 60S fractions appears very similar in both cases.

- h) Recently published cryoEM structures captured translation factors in action on ribosomes. Maybe the authors want to compare their MD simulations with these structures in order to strengthen their conclusions?
- i) The Discussion is way too long and should be trimmed and focused.
- j) Page 24: The sentence "The CTD phosphorylation not only optimizes the ribosome performance in translation per se to meet cell expectations, but can also trigger a functional expansion of ribosomal properties, extending the ribosomal interactome beyond the very specific group of translational factors, such as the Gcn2 kinase." is actually not supported by their own data. The authors demonstrate that Gcn2 binds to ribosomes even when the P-stalk is de-phosphorylated.

Minor points:

- a) The authors should be consistent in the text: "P-stalk protein" not "P-protein"
- b) Page 6: "MilliQ" water and not "MiliQ"
- c) Page 17: "ribosome-protected footprint" is abbreviated with RPF (and not RFP).

Referee #3:

In this study Filipek et al. investigated the functional role of phosphorylation of the oligomeric functional ribosomal proteins, termed P-stalk, whose modification occurs exclusively at the conserved C-terminal regions. The authors constructed the mutant ribosome (and mutant strain) whose P-stalk is in the non-phosphorylated state (SDPM) by serine-to-alanine substitutions within the C-termini of P-stalk proteins. Then, they compared the activities of SDPM ribosomes with those of the wild-type (phosphorylated) ribosomes. Their results showed that the phosphorylation of P-stalk has no significant effect on each step of translation elongation (in vitro) and overall translational efficiency in optimal conditions (in vivo). However, in tunicamycin (Tun)-induced stress conditions for cells, the authors observed some effect of the P-stalk phosphorylation, i.e., the phosphorylation plays a role in timely regulation of the binding of Gcn2 kinase to 60S ribosomes and of its activity in phosphorylation of eIF2 factor, that seems to depend on ribosome stalling on mRNA. From the results of detailed analysis, the authors draw a conclusion that the P-stalk phosphorylation is involved in Gcn2-dependent stress response. All experiments using a wide range of in vivo and in vitro methods with yeast cells and purified factors are well designed. To my knowledge, this study is the first attempt to show the functional role of P-stalk phosphorylation in stress responses. However, there are issues that cannot be overlooked and should be addressed in this research.

The first is the effect of amino acid substitutions in the SDPM ribosome. The authors replaced serine in the CTD of P-stalk with alanine to create a non-phosphorylated form of P-stalk, but did not provide any data on the functional effects of this substitution. Because the effects are not very large in most of the functional data presented in this paper, we cannot exclude the possibility that the observed effects are due to amino acid substitution rather than phosphorylation. Since the authors have succeeded in preparing dephosphorylated ribosomes by alkaline phosphatase treatment (AP-ribosome) as shown in Figs. 2 and 4, the authors should show some kind of in vitro experimental data showing the effect of phosphorylation using wild-type ribosomes (phosphorylated) and AP-ribosomes (dephosphorylated). The authors' conclusion would be most strengthened if the effect of P-stalk phosphorylation on the interaction between Gcn2 and ribosomes, or on the phosphorylation efficiency of eIF2, could be confirmed by in vitro analysis comparing the activities of WT ribosomes and AP-ribosomes.

The second issue is about Figure 10; I feel confused that molecular dynamics (MD) simulation data of human P-stalk CTD, not of yeast P-stalk CTD, is shown as Figure 10. All functional data on P-stalk phosphorylation in this study are from experiments using yeast cells and purified yeast samples. However, the authors show the results of MD simulation of human P-stalk proteins in Figure 10 and discuss about the role of "U-bent shape" in DISCUSSION. As shown in Figures S7A, the amino acid sequences of the N-terminal part of the CTD of yeast P-stalk proteins are different from those of human P-stalk proteins, and the clear U-bent shape discussed in human P-stalk proteins is not seen in yeast P-stalk proteins (S7E). Because the authors have performed careful functional analysis of the phosphorylation of yeast P-stalk in this study, it seems appropriate to report the MD analysis of yeast samples (perhaps as Figure 10). I think MD simulations of human samples could be used as supplementary data.

Minor comments

1. Page 3, line 17: change a phrase "P-stalk forms pentamers uL10(aP1)4 or uL10(P1-P2)2, respectively" to "P-stalk forms heptamers, uL10(aP1)6, or a pentamers, uL10(P1-P2)2", respectively".
2. Page 4, lines 7-9: "although the phosphorylation of P-proteins was discovered more than five decades ago, the phospho-status of the stalk proteins and the physiological role of this modification have not been explored." Here, the authors should take into account the content of a previous report showing that rat phosphorylated P-stalk protein promotes interactions with eEF-2 (J. Biol. Chem. 272, 20259-20262, 1997).
3. Page 11, line 7: It should be stated which mRNA is used.
4. Page 15, line 20: "phosphorylated P-stalk showed a significantly lower affinity toward eIF5B". Because the difference between the dissociation constants $K_d = 4.6 \pm 0.8 \mu\text{M}$ and $11.2 \pm 2 \mu\text{M}$ is not so significant, the word "significantly" should be changed appropriately.
5. Page 16: There is no description of "(fig. 6B)" in text. It should probably be written around line 18 of page 16.

6. Page 16, lines 20-21: "including the activity of eEF2 and eEF3" Did the authors use eEF3 in this study?
7. Figure 2-4: The difference in molecular weight between animal P-stalk proteins P1 and P2 is small, and both proteins are phosphorylated, but P2 is missing in the Phos-tag/Western Blotting in Fig. 2-4. The authors need some kind of excuse/explanation.
8. Figure 9: In immunoblotting pattern shown in the lower right panels of Fig. 9B, the relative amounts of Gcn2 bound to the 60S subunit of the SDPM ribosome appears to be clearly different between Tun+ and Tun-, but in the bar graph (Fig. 9C), the amounts for Tun+ and Tun- appear to be equivalent. The authors need a convincing explanation or a modification of the figure to address this discrepancy.
9. Figure 9 legend: There is no notation of "C" in Fig. 9 legend.

Point-by-point response to the concerns of the referee

Referee #1

In this manuscript Filipek et al use a combination of growth assays, biochemical analyses, translation assays and molecular dynamics simulations to decipher a role for phosphorylation of the stalk proteins. The work is carefully executed and of interest. The data support the model. I have a few suggestions to improve the impact of the work, and increase confidence in the conclusions.

1. The differences in binding of eIF5B to the phosphorylated and dephosphorylated stalk is likely an underestimate, because some of the phosphorylated stalk is dephosphorylated. The authors should note this.

- The reviewer's comment is correct. Based on our analyses shown in Supplementary Figure S2 (current Fig. EV2), a fraction of the stalk is indeed not fully phosphorylated. According to native-MS analysis, up to four phosphate groups are incorporated, so we do not observe the P-stalk fully saturated with phosphate groups. Phos-tag analysis showed that the uL10 comprises a mixture of phosphorylated and dephosphorylated species, whereas P1/P2 are more homogeneous in terms of their phosphorylation status. We agree that it may influence the K_d value. Although the issue of the mixed phospho/dephospho population was mentioned in the text, we have changed it based on the reviewer's comment to give readers more clarity on the samples used.

2. It would be great to see at least an SDS-Page gel (or better 2D gel, or mass-spec data) to confirm that the wt and SDMP ribosomes are otherwise identical.

- We provide the MS data of the purified ribosomes in the supplementary data as Appendix Table S1. Information has been added to the Materials and Methods section and also to the Results section to provide the reader with information on the quality of the sample preparation.

3. The authors carry out a very systematic analysis of effects arising from the phosphor-deficient SDMP mutant. I'd be curious to know if the phosphomimetic mutant has any phenotypes, especially under stress, or conditions that lead to open A-sites.

- This comment raises an interesting point about generating yeast mutants by replacing serine residues with phosphomimetic mutations, such as glutamic acid or aspartic acid. This modification could render the yeast strain resistant to dephosphorylation, which in turn could indeed affect the yeast strain's response to environmental changes. The reviewer's concern is valid, as this modification could affect the behavior of the yeast under different conditions. First, we did not observe any fluctuations in the phosphorylation status of the P-proteins when yeast or mammalian cell lines were exposed to different stress conditions, suggesting that this status represents the standard state of the P-stalk protein. In a second experiments, we chose to evaluate a mutant in which all five serine residues in the P-stalk proteins were replaced by alanine – SDPM mutant strain, to obtain the yeast strain most different from the wild-type, which is permanently phosphorylated. It should be noted, that the SDPM mutant stain, in basal conditions, had similar characteristics as the WT strain, however in upon stress application (tunicamycin treatment) showed lack of the correct adaptation to adverse

conditions, indicating that the presence of phosphate group represent the optimal state for the P-stalk proteins. We also analyzed the dephosphorylation kinetics of the P-stalk proteins upon application of a CK2 kinase inhibitor. The results indicate that the process is very slow and is not influenced by any specific phosphatase action (data not presented in the manuscript).

4. I would suggest the authors view everything in relation to the wt strain. E.g. p. 18, l. 10: the codon occupancy was decreased in the mutant...

- We agree, that the information about WT and SDPM strains under control conditions was not accurately referenced in Fig. 8B, left panel. The text has been adjusted to clearly refer to the control conditions for both WT and SDPM strains. We believe this correction provides greater clarity in the data presentation.

5. The suggestions from the MD should be tested by mutagenesis of the lysine residues. Their effects can be tested in the in vivo assays that give a phenotype for the SDPM mutant.

- Yes, the reviewer is correct that the role of lysine residues can be tested experimentally; however, this would require the generation of a yeast mutant strain in which all lysine residues in all five P-stalk proteins are mutated, which is a very time-consuming approach, as was the case for the mutant strain with S to A mutations. However, taking into account the reviewer's suggestion, we have performed the MD simulation of lysine-to-alanine mutated forms of the P-stalk C-terminal peptides (considering the P-proteins that have the lysine residues – human P1, P2, uL10, and yeast P1A and P2B). The analysis showed that the peptides are unable to form the bends and turns responsible for the U-bend conformation in the N-terminal part of the peptide, suggesting that lysine residues play a crucial role in stabilizing the U-bend conformation. We have added new data panel within the Appendix section, as Fig. S3N.

Referee #2:

In the submission by Filipek et al., the authors test the hypothesis that differential phosphorylation levels of the eukaryal P-stalk protein uL10 regulates and possibly interconnects both the ribosome's translational activity (by affecting recruitment/activation of translational GTPases) with the Integrated Stress Response (by activation of the stress-related Gcn2 kinase). Given their major findings that P-stalk proteins are exclusively in the phosphorylated state and that the phosphorylation level of the P-stalk is constant and the fact that protein biosynthesis activities of the translational machinery are largely unaffected, the hypothesis does not seem to be supported by the obtained data.

The presented data are (most of the time) of high quality and are thus relevant for falsifying the starting hypothesis. There are several points that merit consideration:

a) Introduction, page 3: The authors should mention that the ribosomal P-stalk architecture also involves uL11 (beside uL10). It has been shown in yeast and human system that uL11 contributes to translation dynamics.

- We agree and have incorporated this information into the Introduction to highlight the involvement of uL11 in the P-stalk architecture and function.

b) Fig. 4A: The authors state in the Results that "The AP treatment of the isolated cell extracts prior to fractionation did not affect the overall shape of polysomal profiles in the yeast (Fig. 4A, left) or mammalian (Fig. 4A, right) cells." This statement is not supported by the polysome profile in mammalian cells since especially the heavy polysomes were significantly altered upon AP treatment.

- Yes, this is indeed the case. The fraction of heavy polysomes, especially in yeast polysomes, decreased after AP treatment. As the original description in the text did not sufficiently address this issue, we have modified the text to provide greater clarity and to explain the behavior of the polysomal fractions analyzed. We would like to add that the observed behavior can be explained by the fact that the polysomal fraction was incubated with AP, allowing some ribosomes time to complete the translation. As a result, the heavy polysome fraction decreased. Further, according to our observations, yeast polysomes have a high tendency to complete translation during longer incubation times (known as run-off analysis), whereas ribosomes from mammalian cell lines have a lower tendency to do so. However, we did not observe any changes in the phospho-status of the P-stalk proteins upon prolonged polysome preparation.

c) Fig. 4B: The quality of the Phos-tag western blot for the yeast sample is quite poor which affects the conclusions that the authors draw from these experimental data. e.g. the bands in the last fraction #9 are half cut.

- The mistake occurred during the figure preparation, resulting in the data being partially cut, as noted by the reviewer. We apologize for this error. The figure has been revised to display the entire last fraction correctly. We agree that the quality of the yeast samples analyzed by Phos-tag western blot is not perfect. However, this is primarily due to the quality of the antibodies, particularly those targeting yeast P1/P2 proteins. There are no high-quality antibodies available on the market to detect yeast P1/P2 proteins (specifically, P1A, P1B, P2A, P2B). We used antibodies prepared in-house based on purified recombinant yeast P1/P2 proteins, but this is the maximum quality that we can achieve with the existing reagents.

d) Fig. S3: it has been shown before that the dual luciferase reporter used here, which relies on the misincorporation of amino acids at the His245 codon, is not ideal in the yeast system due to its very high background activity for many amino acids incorporated at that position within the active site. It has been demonstrated that the Lys529 mutant is superior due to its low background signal.

- The reviewer's concern is related to the ongoing debate about the reliability of data based on the dual luciferase reporter system, particularly with regard to misincorporation. We agree that the use of the CGC245 codon for the amino acids His245 is not ideal. Bearing in mind the limitations, we have also used an additional reporter system based on AGC218 to support our analysis using the CGC245 codon. The data from both systems are consistent and we believe that the use of two reporters meets the requirements for obtaining reliable data. All data were presented in Fig. S3 (currently, Fig. EV3).

e) Given their main findings (phosphorylation status of the P-stalk is unaltered and has little or no effect on growth as well as on the overall performance of the translational machinery; e.g. first sentence on page 16; or abstract, etc), the wording "Phosphorylation...modulates ribosome functions" or even the title of the manuscript "Phosphorylationfine-tunes ribosomal-factor interactions" is misleading. A reader could anticipate that words such as "fine-tuning" or "modulation" would imply that altered in vivo levels of P-stalk protein phosphorylation would adjust ribosomal activities. The authors however demonstrate in this study that both are actually not happening. Thus I strongly suggest re-considering such statements.

- We agree with the reviewer's comment and the text has been refined accordingly to provide the reader with a clear statement that the P-proteins are exclusively in the phosphorylated state, which significantly distinguishes them from other ribosomal proteins that are transiently phosphorylated and function in an on/off manner, exerting a classical regulatory role. Therefore, the title of the manuscript was modified (current version: Phosphorylation of P-stalk proteins defines the ribosomal basal state for interaction with auxiliary protein factors). Equivalent changes were also introduced in the abstract and the main text.

f) The effects of tunicamycin are typically seen after several hours (once the ER gets too crowded and the UPR gets activated). The observed response after solely 15 min but not after 60 min is therefore surprising (Fig. 8A). Are cells already stressed after 15 min tunicamycin treatment? Could the authors use e.g. eIF2alpha phosphorylation or activation of any of the known ER-stress sensors to convince the reader of an indeed activated stress response?

- We agree that the tunicamycin (Tun) effect typically develops gradually and can be observed several hours after tunicamycin addition in higher eukaryotic cells, such as in mammalian cell lines. Specifically, following several markers, such as XBP1 mRNA splicing and activation of PERK and phosphorylation of eIF2 α , the activation of the UPR is slow and requires time in mammals. As it proposed, the capacity of the ER lumen in mammalian cells may function as a buffering compartment with a significantly high capacity for unfolded proteins, thereby delaying the activation of XBP1 mRNA splicing and the late activation of the UPR. In contrast, in yeast, the ER lumen is less developed and has a significantly lower capacity. Several reports have shown that splicing of HAC1 (the homologous protein to mammalian XBP1) occurs much more rapidly, even one hour after treatment with Tun. Based on our preliminary data, we noticed that Tun had an immediate effect in yeast

cells. Therefore, we chose 15 minutes as the first time point to follow ribosome behavior on mRNA using RIBOseq. Following the reviewer's suggestion, we are providing our analysis showing HAC1 splicing, using the classical PCR approach to show the time course of unspliced/spliced forms of HAC1 mRNA. The analysis showed that the spliced form of HAC1 mRNA is already present after 15 minutes of Tun treatment, indicating that the UPR is initiated very rapidly in yeast cells. The supplementary data have been added as Appendix Figure S2 and the text in the Results section has been updated to reflect the newly presented information.

- Additionally, reviewer raised the issue related to eIF2 α phosphorylation. We are providing the response below.

The data shown in Fig. 9A do not appear to suggest stress activation after 15 min Tun treatment. This should be clarified.

- The eIF2 α phosphorylation observed upon Tun treatment showed a one-hour delay compared to our RIBOseq data, which indicated ribosome stalling on CCG, CGG, and CGA codons just 15 minutes after Tun treatment. In our opinion, the RIBOseq data illustrate the initiation of the stress response, where ribosomes act as sensors by stalling on the mRNA (particularly at Arg and Pro codons). This stalling signals downstream through the Gcn2 kinase, which subsequently phosphorylates eIF2 α . This suggests a delay of about one hour between the initial sensing event and the resulting effect. Based on the reviewer's remark, we conclude that this issue was not sufficiently explained in the original text. In the discussion section, we now provide readers with a more comprehensive interpretation, incorporating recent findings that indicate the signal transmission from the ribosome to downstream effectors functions in a rheostat-like manner (see reference 92). This expanded explanation addresses the point raised by the reviewer and is in line with the latest insights on how this buffering mechanism may operate.

g) The authors state that Gcn2 accumulates on the 60S ribosomal subunit in the WT strain upon Tun treatment, but not in the SDPM cells. Based on the blot shown in Fig. 9B such a statement does not seem to be justified given the fact that the western blot signal for Gcn2 in the 60S fractions appears very similar in both cases.

- The reviewer has raised a crucial point regarding the activation mechanisms of Gcn2, which is a topic of ongoing debate. The question of how Gcn2 is activated within the cell is not yet fully resolved. Current models suggest that Gcn2 can be activated either by tRNA in the cytoplasm or by stalled/collided ribosomes. Given that the P-stalk plays a role in Gcn2 activation, we have taken the approach of examining the interplay between Gcn2 and ribosomes, including analyzing polysome fractions, to shed more light on this unresolved issue. While the data presented in Fig. 9B, particularly the western blotting results, may appear difficult to interpret at first glance, it is important to look beyond a visual assessment. Although a general visual inspection might not immediately reveal clear trends, a detailed analysis of signal intensity relative to the control samples provides accurate numerical data. This quantification supports the validity of our findings and strengthens the argument for the ribosome-related activation mechanisms of Gcn2 as demonstrated through our western blotting analysis. To our knowledge, this is the first comprehensive biochemical trial to define Gcn2 association with ribosomes, which may open the window for further investigations.

h) Recently published cryoEM structures captured translation factors in action on ribosomes. Maybe the authors want to compare their MD simulations with these structures in order to strengthen their conclusions?

- We agree with the reviewer that recently published cryo-EM structures of ribosomes, with a particular focus on stalled/collided ribosomes, have provided a significant step forward in the understanding of the structural nature of ribosomes as stress 'sentinels', and in a majority of them the P-stalk represents an important player. To mention just the most important: the work presented by Danel Wilson's laboratory (*Structure of Gcn1 bound to stalled and colliding 80S ribosomes, PNAS, 2021, 118(14):e2022756118*) has laid the foundation for understanding ribosome collisions at the atomic level. Recent follow-up work by Friedrich Förster's laboratory has shown the ensemble of ribosomal structures under stress conditions, in complex with translation factors (tRNA-eEF1A, eEF2) and with the Gcn1/Gcn20 disome complex, and has also significantly expanded our knowledge of ribosome collisions, in particular distinguishing colliding disomes from compact helical polysomes, which appear to reflect severe stress (*Visualization of translation reorganization upon persistent ribosome collision stress in mammalian cells, Mol Cell, 2024, 84(6):1078-1089.e4*). Or the very recent elegant work from Andrei Korostelev's lab, where the authors showed in *Nature, 2024, 630(8017):769-776, 'Structural mechanism of angiogenin activation by the ribosome'*, that the ribosome with an empty A site can accommodate angiogenin and specifically process the tRNA; importantly, these authors provide a new concept, stating that: “*This structural mechanism accounts not only for the activation of angiogenin, but also for the substrate specificity provided by the ribosome, of which the P stalk is tuned for tRNA delivery*” suggesting, that the flexible part of the P-stalk may act as an anchoring part for the free tRNA, expanding the repertoire of P-stalk partners. However, in the light of the above and other cryo-EM structural work, the flexible part of the P-stalk is missing in all available structures, and in particular the C-terminal end of the P-stalk protein has never been resolved; as such, we only have insight into the N-terminal part of the uL10 and P1/P2 proteins, which is seen as the protruding part. Thus, although aligning the experimental cryo-EM data with our MD simulation would be desirable, the cryo-EM data are not complete enough to make this feasible.

i) The Discussion is way too long and should be trimmed and focused.

j) Page 24: The sentence "The CTD phosphorylation not only optimizes the ribosome performance in translation per se to meet cell expectations, but can also trigger a functional expansion of ribosomal properties, extending the ribosomal interactome beyond the very specific group of translational factors, such as the Gcn2 kinase." is actually not supported by their own data. The authors demonstrate that Gcn2 binds to ribosomes even when the P-stalk is de-phosphorylated.

- The text has been improved and shortened for clarity.

Minor points:

a) The authors should be consistent in the text: "P-stalk protein" not "P-protein"

b) Page 6: "MilliQ" water and not "MiliQ"

c) Page 17: "ribosome-protected footprint" is abbreviated with RPF (and not RFP).

- The corrections were introduced

Referee #3:

In this study Filipek et al. investigated the functional role of phosphorylation of the oligomeric functional ribosomal proteins, termed P-stalk, whose modification occurs exclusively at the conserved C-terminal regions. The authors constructed the mutant ribosome (and mutant strain) whose P-stalk is in the non-phosphorylated state (SDPM) by serine-to-alanine substitutions within the C-termini of P-stalk proteins. Then, they compared the activities of SDPM ribosomes with those of the wild-type (phosphorylated) ribosomes. Their results showed that the phosphorylation of P-stalk has no significant effect on each step of translation elongation (in vitro) and overall translational efficiency in optimal conditions (in vivo). However, in tunicamycin (Tun)-induced stress conditions for cells, the authors observed some effect of the P-stalk phosphorylation, i.e., the phosphorylation plays a role in timely regulation of the binding of Gcn2 kinase to 60S ribosomes and of its activity in phosphorylation of eIF2 factor, that seems to depend on ribosome stalling on mRNA. From the results of detailed analysis, the authors draw a conclusion that the P-stalk phosphorylation is involved in Gcn2-dependent stress response. All experiments using a wide range of in vivo and in vitro methods with yeast cells and purified factors are well designed. To my knowledge, this study is the first attempt to show the functional role of P-stalk phosphorylation in stress responses. However, there are issues that cannot be overlooked and should be addressed in this research.

The first is the effect of amino acid substitutions in the SDPM ribosome. The authors replaced serine in the CTD of P-stalk with alanine to create a non-phosphorylated form of P-stalk, but did not provide any data on the functional effects of this substitution. Because the effects are not very large in most of the functional data presented in this paper, we cannot exclude the possibility that the observed effects are due to amino acid substitution rather than phosphorylation.

- Indeed, in the manuscript we show a few phenotypic analyses indicating that there is a small but statistically significant defect in SDPM growth. The classical growth rate analysis showed that the absence of P-stalk phosphorylation had a small, but statistically significant negative effect on cell growth. These data were further supported by single-cell level analysis, which evaluated various aspects of yeast lifespan and showed that the SDPM mutant strain had similar characteristics as the WT strain, albeit with a reduced reproductive lifespan. Although we cannot exclude the possibility that under other conditions the metabolic discrepancy may be more pronounced, we were not able to identify such conditions at the current stage of this study.

Since the authors have succeeded in preparing dephosphorylated ribosomes by alkaline phosphatase treatment (AP-ribosome) as shown in Figs. 2 and 4, the authors should show some kind of in vitro experimental data showing the effect of phosphorylation using wild-type ribosomes (phosphorylated) and AP-ribosomes (dephosphorylated).

- We agree, and we have indeed performed the experiments with AP-treated ribosomes (dephosphorylated), as noted in Fig. 2 and 4, and also in the MST analysis, where we used dephosphorylated ribosomal P-stalk complex (Fig. 5A). However, in the case of the *in vitro* GTP hydrolysis assay and *in vitro* translation analysis, we could not use AP-ribosomes, because the residual presence of AP interfered with the analyses.

The authors' conclusion would be most strengthened if the effect of P-stalk phosphorylation on the interaction between Gcn2 and ribosomes, or on the phosphorylation efficiency of eIF2, could be confirmed by in vitro analysis comparing the activities of WT ribosomes and AP-ribosomes.

- The reviewer's comment is in line with the core experiments presented in the manuscript and refers to the *in vitro* analysis that could be performed using isolated elements such as Gcn2, ribosomes and eIF2 α . Regarding the *in vitro* analysis to study the interaction between Gcn2 and ribosomes, including eIF2 α phosphorylation, there are several significant experimental challenges. The problem relates to the interaction between

Gcn2 and the ribosome, which is currently a challenge in the field. Previous reports have shown that the interaction between Gcn2 and the ribosome is complex and not fully understood, particularly under stress conditions. This interaction may involve other proteins, including Gcn1 and Gcn20. In addition, there is a debate as to which form of ribosome is the 'substrate' for Gcn2, 60S, 80S or dimers/disomes 80S-80S, in the form of so-called collided ribosomes. Therefore, we believe that an *in vitro* analysis attempting to reconstruct the interaction of Gcn2 with purified ribosomes will be a large project on its own and will be addressed in future research.

- Another issue raised by the reviewer concerns the *in vitro* experiment of eIF2 α phosphorylation taking into account ribosomes and Gcn2 kinase. As stated above, purification of Gcn2 kinase from yeast cells represents a major challenge, as the kinase is purified in its active form, and this complicates the assay and makes these experiments unfeasible. It should be added that a similar experiment has already been performed, but in a mammalian experimental model, using purified eIF2 α , Gcn2 kinase and ribosomes, as well as the pentameric P-stalk complex, but the phosphorylation of the P-stalk proteins was not taken into account and not tested (*Proc. Natl. Acad. Sci. USA.* 2019,116(11):4946-4954). We would like to admit, that we have not yet developed the complex experimental tools to perform such analysis in yeast or in mammals and argue that currently this is beyond the scope of the current manuscript.

The second issue is about Figure 10; I feel confused that molecular dynamics (MD) simulation data of human P-stalk CTD, not of yeast P-stalk CTD, is shown as Figure 10. All functional data on P-stalk phosphorylation in this study are from experiments using yeast cells and purified yeast samples. However, the authors show the results of MD simulation of human P-stalk proteins in Figure 10 and discuss about the role of "U-bent shape" in DISCUSSION. As shown in Figures S7A, the amino acid sequences of the N-terminal part of the CTD of yeast P-stalk proteins are different from those of human P-stalk proteins, and the clear U-bent shape discussed in human P-stalk proteins is not seen in yeast P-stalk proteins (S7E). Because the authors have performed careful functional analysis of the phosphorylation of yeast P-stalk in this study, it seems appropriate to report the MD analysis of yeast samples (perhaps as Figure 10). I think MD simulations of human samples could be used as supplementary data.

- We showed the human data in Fig. 10 because we wanted to show the most prominent example of the P-stalk protein phosphorylation effect. We agree that in most cases the publication is related to the yeast model, and in order to provide the reader with a complete picture, we have modified Fig. 10 to show all models, including mammalian, yeast and archaeal structures.

Minor comments

1. Page 3, line 17: change a phrase "P-stalk forms pentamers uL10(aP1)4 or uL10(P1-P2)2, respectively" to "P-stalk forms heptamers, uL10(aP1)6, or a pentamers, uL10(P1-P2)2", respectively".

- Corrected

*2. Page 4, lines 7-9: "although the phosphorylation of P-proteins was discovered more than five decades ago, the phospho-status of the stalk proteins and the physiological role of this modification have not been explored." Here, the authors should take into account the content of a previous report showing that rat phosphorylated P-stalk protein promotes interactions with eEF-2 (*J. Biol. Chem.* 272, 20259-20262, 1997).*

- New text was provided in the introduction and discussion sections to take into account the information provided in the cited publication.

3. Page 11, line 7: It should be stated which mRNA is used.

- The mRNA used for the *in vitro* translation assay was added into Materials and Methods section.

4. Page 15, line 20: "phosphorylated P-stalk showed a significantly lower affinity toward eIF5B". Because the difference between the dissociation constants $K_d = 4.6 \pm 0.8 \mu\text{M}$ and $11.2 \pm 2 \mu\text{M}$ is not so significant, the word "significantly" should be changed appropriately.

- Following the reviewer suggestion, we have removed the expression "significant" and the new description was added stating that: "... the native phosphorylated P-stalk showed a two times lower affinity toward eIF5B than the dephosphorylated one."

5. Page 16: There is no description of "(fig. 6B)" in text. It should probably be written around line 18 of page 16.

- The missing description of Fig. 6B was implemented into the text describing tripeptide formation analysis.

6. Page 16, lines 20-21: "including the activity of eEF2 and eEF3" Did the authors use eEF3 in this study?

- Yes, for the tripeptide analysis, eEF2 and eEF3 were used to initiate the translocation. The Materials and Methods section has been updated to provide detailed information.

7. Figure 2-4: The difference in molecular weight between animal P-stalk proteins P1 and P2 is small, and both proteins are phosphorylated, but P2 is missing in the Phos-tag/Western Blotting in Fig. 2-4. The authors need some kind of excuse/explanation.

- We apologize for not providing a thorough explanation for the lack of P2 detection. In short, the available antibodies against animal P1/P2 proteins were of variable quality, and the only antibodies that gave us reliable detection were antibodies against P1 protein, while anti-P2 antibodies were of variable quality in terms of reliable detection of the protein. Since, P1-P2 form a functional dimer on the ribosome, we decided to use only anti-P1 as a readout for the P1/P2 proteins. The explanation was provided in the Materials and Method section.

8. Figure 9: In immunoblotting pattern shown in the lower right panels of Fig. 9B, the relative amounts of Gcn2 bound to the 60S subunit of the SDPM ribosome appears to be clearly different between Tun+ and Tun-, but in the bar graph (Fig. 9C), the amounts for Tun+ and Tun- appear to be equivalent. The authors need a convincing explanation or a modification of the figure to address this discrepancy.

- We acknowledge the reviewer's concern regarding the interpretation challenges of the western blotting data in Figure 9B. Indeed, the visual inspection alone may not clearly show the differences. In the provided representative western blot, the Gcn2 protein levels appear to vary between Tun -/+ conditions. However, when normalized against the reference protein uL10, the Gcn2 levels are comparable. We recognize that the observed variations, indicated by the substantial error bars in Figure 9C, contribute to our classification of these differences as non-significant. However, we would like to emphasize that analyzing Gcn2 presents a significant challenge due to the limited methodologies available to demonstrate

its association with the ribosomal fraction. Consequently, we believe that our data offer valuable insights for understanding Gcn2 behavior, addressing an issue that has not yet been resolved. We believe that our findings contribute to the broader knowledge and will aid in advancing the field's understanding of Gcn2's role and interactions with ribosomes. To enhance clarity for our readers, we have revised the text to place a stronger emphasis on the data processing methods used. We would like to add that the isolation of Gcn2 kinase in complex with ribosomes is of great interest to us and the methodology is under development.

9. Figure 9 legend: There is no notation of "C" in Fig. 9 legend.

- The text was corrected.

Dear Prof. Tchorzewski,

Thank you for the submission of your revised manuscript. We have now received the enclosed reports from the referees. As you will see, referees 2 and 3 still have some remaining concerns that I would like you to address before we can proceed with the official acceptance of your manuscript.

Both referees are not convinced by the data shown in Figs 9B and C. These data are based on 2 independent experiments. I would like to suggest that you repeat the experiment at least one more time so that $n=3$. Please do not add up the technical repeats with the experimental repeats, as is currently done. Hopefully, comparing the outcome of at least 3 independent experiments will yield clearer results. Please let me know in case you disagree, and we can discuss this further. I do not agree with referee 3 that any obtained data should be excluded from the figure or ms.

A few other editorial requests will also need to be addressed:

- Please add put to 5 keywords to the ms file.
- Please correct the conflict of interest subheading to "Disclosure Statement and Competing Interests"
- Please correct these name discrepancies: Colin Chih-Chien Wu in the ms file vs. Colin Wu in our online submission system (eJP); Alan González-Ibarra in the ms file vs. Alan González in eJP; Yutian Shao in the ms file vs. Shao Yutian in eJP; all corresponding authors' emails need to be provided on the title page.
- Please remove the author credits from the ms file. All credits need to be entered during online ms submission.
- The REFERENCE FORMAT needs to be corrected to EMBO reports (Harvard) style. It needs to be alphabetical (not numerical); et al needs to be used after 10 names.
- Please remove DATA NOT SHOWN on p14 and p22, as per journal policy. Please either show the data or re-phrase.
- Please complete all questions in the statistics section of the author checklist and send us a fully completed checklist. It will also be part of your transparent peer-review process file.
- Please enter the missing ORCID IDs for Cunningham and Rodnina on their personal profile pages.
- There are 2 Appendix PDF files - one with figures, the other with tables. These need to be merged into one PDF; "Table S1" should be corrected to "Appendix Table S1"; the one Appendix PDF needs to have a title page with a short table of content with page numbers; missing "Appendix" word in the title of Appendix Figure S3; Appendix Figure S3 runs over many pages. It would be better to make each panel a new Appendix Figure: Appendix Figure S4, etc.; on p11 there is a Figure S8 label that needs to be corrected.
- Please remove the Instructions from the Reagents and Tools Table.
- Please upload the source data as one folder per figure.
- Materials and Methods should be Methods
- The manuscript sections should be in the following order: Title page - Abstract & Keywords - Introduction - Results - Discussion - Methods - Data Availability - Acknowledgments - Disclosure Statement & Competing Interests - References - Figure Legends - (Main Tables with legends) - Expanded View Figure Legends.
- Please note that the exact p values are not provided in the legends of figures 9a, c; EV 3; EV 4a-b.
- Please note that information related to n is missing in the legends of figures 5b-d.
- Please note that the error bars are not defined in the legends of figures 5b-d; EV 4a.
- Please note that the measure of center for the error bars needs to be defined in the legends of figures 9a, c; EV 3.
- Please note that axis labels are not defined for figures EV 3.

EMBO press papers are accompanied online by A) a short (1-2 sentences) summary of the findings and their significance, B) 2-

3 bullet points highlighting key results and C) a synopsis image that is exactly 550 pixels wide and 200-600 pixels high (the height is variable). The synopsis image should provide a sketch of the major findings, like a graphical abstract. Please note that text needs to be readable at the final size. Please send us this information along with the final manuscript.

I look forward to seeing a new revised version of your manuscript as soon as possible.

Referee #1:

This revised manuscript addresses my previous concerns.

Referee #2:

In the revised submission by Filipek et al., the authors addressed all the points raised. While some of the critical points were convincingly clarified [a, c, d, e, f (the HAC splicing part), h], other points remain not completely settled [b, f (the eIF2alpha part: even after 2 h tunicamycin treatment no Gcn2 enrichment in polysomes is observed, Fig. 9, which makes it hard to follow the author's logic about the one-hour delay of Gcn2-mediated eIF2alpha phosphorylation: The sentence in their rebuttal letter "While the data presented in Fig. 9B, particularly the western blotting results, may appear difficult to interpret at first glance, it is important to look beyond a visual assessment." does not sound terribly convincing to me about the robustness of the data and the analysis.)].

Overall I appreciate the authors' efforts to accommodate mine (and the two other reviewer's) concerns. As a result, the revised version has improved.

Referee #3:

The study by Filipek et al. seems to be the first attempt to show the functional role of phosphorylation of P-proteins in stress responses. However, I had some concerns about their original manuscript and I offered some suggested revisions. In response to my comments, the authors have made appropriate revisions, with the following exceptions.

In response to my first comment about the functional effect of serine-to-alanine substitutions within the C-termini of P-proteins, the authors have responded that it would be difficult to conduct experiments to address my comment under the current circumstances. I understand that such experiments are currently difficult, and beyond the scope of this manuscript.

For my minor comment #8: The authors have explained the data, showing the representative western blot (Fig. 9B) and band intensities normalized against the band of L10 (Fig. 9C). The authors have also recognized the observed variations of data, as indicated by large error bars (Fig. 9C). I still have concerns about the striking difference between the western blot results (Fig. 9B) and the quantitative data shown in the bar graph (Fig. 9C) regarding the amount of Gcn2 bound to the SDPM ribosome 60S subunit under stress conditions. Referee #2 also expressed a similar concern in a comment (g).

I would like to offer a further suggestion for amending the manuscript. How about replacing the panels in Fig. 9B with more suitable ones, if possible? Otherwise, how about excluding Fig. 9C and interpreting Fig. 9B as meaning that Gcn2 has potential ability to bind to the ribosomal 60S subunit, that is strengthened by the Tun-induced stress, and that phosphorylation of the P-protein further strengthens this binding? In the case of the latter correction, it will be necessary to revise some of the expressions in the text accordingly.

A newly found error

Page 10, line 1: "VEAKEESEDEDMGFGLFD peptide was used for the human protein uL10" The correct sequence may be "VEAKEESESEDEDMGFGLFD".

Point-by-point response to the concerns of the referee

We wish to thank the reviewers for their detailed assessment of the manuscript. Their valuable comments and helpful suggestions have significantly improved the data presentation and the quality of the present work. We are grateful for their time and expertise, which contributed greatly to the improvement of our work.

Referee #1:

This revised manuscript addresses my previous concerns.

- Thank you for your valuable remarks.

Referee #2:

In the revised submission by Filipek et al., the authors addressed all the points raised. While some of the critical points were convincingly clarified [a, c, d, e, f (the HAC splicing part), h], other points remain not completely settled [b, f (the eIF2alpha part: even after 2 h tunicamycin treatment no Gcn2 enrichment in polysomes is observed, Fig. 9, which makes it hard to follow the author's logic about the one-hour delay of Gcn2-mediated eIF2alpha phosphorylation: The sentence in their rebuttal letter "While the data presented in Fig. 9B, particularly the western blotting results, may appear difficult to interpret at first glance, it is important to look beyond a visual assessment." does not sound terribly convincing to me about the robustness of the data and the analysis.)]. Overall I appreciate the authors' efforts to accommodate mine (and the two other reviewer's) concerns. As a result, the revised version has improved.

- Response below

Referee #3:

The study by Filipek et al. seems to be the first attempt to show the functional role of phosphorylation of P-proteins in stress responses. However, I had some concerns about their original manuscript and I offered some suggested revisions. In response to my comments, the authors have made appropriate revisions, with the following exceptions.

In response to my first comment about the functional effect of serine-to-alanine substitutions within the C-termini of P-proteins, the authors have responded that it would be difficult to conduct experiments to address my comment under the current circumstances. I understand that such experiments are currently difficult, and beyond the scope of this manuscript.

For my minor comment #8: The authors have explained the data, showing the representative western blot (Fig. 9B) and band intensities normalized against the band of L10 (Fig. 9C). The authors have also recognized the observed variations of data, as indicated by large error bars (Fig. 9C). I still have concerns about the striking difference between the western blot results (Fig. 9B) and the quantitative data shown in the bar graph (Fig. 9C) regarding the amount of Gcn2 bound to the SDPM ribosome 60S subunit under stress conditions. Referee #2 also expressed a similar concern in a comment (g).

I would like to offer a further suggestion for amending the manuscript. How about replacing the panels in Fig. 9B with more suitable ones, if possible? Otherwise, how about excluding Fig. 9C and interpreting Fig. 9B as meaning that Gcn2 has potential ability to bind to the ribosomal 60S subunit, that is strengthened by the Tun-induced stress, and that phosphorylation of the P-protein further strengthens this binding? In the case of the latter correction, it will be necessary to revise some of the expressions in the text accordingly.

- Response below

➤ *En bloc* response to feedback from two reviewers

We recognize that the reviewers concerns are entirely valid and deserve careful consideration, as the mechanism of Gcn2 activation by ribosomal particles remains an unresolved issue. In response to these remarks, we have performed additional experiments to enhance the credibility of our analysis, presented in Fig. 9B. Wild-type and SDPM yeast cells were treated with tunicamycin (Tun) for 2 hours. Polysome fractions were then collected, and the distribution of Gcn2 kinase was assessed by Western blot analysis. The fractions from untreated (-Tun) and treated (+Tun) samples were loaded side by side for better comparison. The obtained data support the previously observed trend of Gcn2 binding to the ribosomal 60S subunit, which is particularly augmented under Tun-induced stress, and phosphorylation of the P-protein increases Gcn2 binding. However, we must admit that quantitative densitometric analysis represents is a significant challenge and also the low resolution of polysome fractionation, which results in fluctuations in fraction collections. Consequently, we were unable to reduce the error rate in the data sets. Based on our recent analyses, we believe that the current approach does not allow a precise quantitative assessment of the Gcn2 content on the ribosomal particles, and that a substantial amount of research needs to be initiated to resolve this issue.

Therefore, to avoid quantitative uncertainty, following reviewer #3 suggestion, we decided to remove Fig. 9C, and we are presenting new Westen blotting, that we believe provides sufficient representation of the observed Gcn2 binding. Taking into account the reviewer's suggestion regarding the manuscript amendment, as the detected Gcn2 binding to the 60S subunit does not represent a binary effect but rather a transient state that is difficult to quantitatively capture with current experimental methods, the manuscript text has been modified to provide a more nuanced perspective regarding Gcn2 binding to the ribosomal subunit.

*A newly found error Page 10, line 1: "VEAKEESDEDMGFGLFD peptide was used for the human protein uL10"
The correct sequence may be "VEAKEESEESDEDMGFGLFD".*

➤ The error was corrected.

Prof. Marek Tchorzewski
Maria Curie-Skłodowska University
Department of Molecular Biology
Akademicka 19
Lublin 20-033
Poland

Dear Prof. Tchorzewski,

I am very pleased to accept your manuscript for publication in the next available issue of EMBO reports. Thank you for your contribution to our journal.

Yours sincerely,
